# Few-layer bismuth selenide cathode for low-temperature quasi-solid-state aqueous zinc metal batteries

Yuwei Zhao[1], Yue Lu[2], Huiping Li[3], Yongbin Zhu[4], You Meng [1], Na Li[1], Donghong Wang[1], Feng Jiang[4], Funian Mo[1], Changbai Long[5], Ying Guo[1], Xinliang Li[1], Zhaodong Huang[1], Qing Li[1], Johnny C. Ho [1], Jun Fan [1], Manling Sui [2], Furong Chen[1], Wenguang Zhu [3✉], Weishu Liu [4✉] & Chunyi Zhi [1,6✉]

The performances of rechargeable batteries are strongly affected by the operating environmental temperature. In particular, low temperatures (e.g., ≤0 °C) are detrimental to efficient cell cycling. To circumvent this issue, we propose a few-layer $Bi_2Se_3$ (a topological insulator) as cathode material for Zn metal batteries. When the few-layer $Bi_2Se_3$ is used in combination with an anti-freeze hydrogel electrolyte, the capacity delivered by the cell at −20 °C and 1 A g$^{-1}$ is 1.3 larger than the capacity at 25 °C for the same specific current. Also, at 0 °C the Zn||few-layer $Bi_2Se_3$ cell shows capacity retention of 94.6% after 2000 cycles at 1 A g$^{-1}$. This behaviour is related to the fact that the Zn-ion uptake in the few-layer $Bi_2Se_3$ is higher at low temperatures, e.g., almost four $Zn^{2+}$ at 25 °C and six $Zn^{2+}$ at −20 °C. We demonstrate that the unusual performance improvements at low temperatures are only achievable with the few-layer $Bi_2Se_3$ rather than bulk $Bi_2Se_3$. We also show that the favourable low-temperature conductivity and ion diffusion capability of few-layer $Bi_2Se_3$ are linked with the presence of topological surface states and weaker lattice vibrations, respectively.

[1] Department of Materials Science and Engineering, City University of Hong Kong, Hong Kong, China. [2] Institute of Microstructure and Properties of Advanced Materials, Beijing University of Technology, Beijing, China. [3] International Center for Quantum Design of Functional Materials (ICQD), Hefei National Laboratory for Physical Sciences at the Microscale, Department of Physics, University of Science and Technology of China, Hefei, China. [4] Department of Materials Science and Engineering, Southern University of Science and Technology, Shenzhen, China. [5] School of Advanced Materials and Nanotechnology, Xidian University, Xi'an, China. [6] Centre for Functional Photonics, City University of Hong Kong, Kowloon, Hong Kong. ✉email: wgzhu@ustc.edu.cn; liuws@sustech.edu.cn; cy.zhi@cityu.edu.hk

Performance loss or failure occurs in nearly all types of cell systems (e.g. lithium/sodium/magnesium/zinc ion cell) in cold climates[1–3]. Advanced batteries imperatively need powerful energy density and excellent lifespan, even at subzero temperatures. Substantial effort has been devoted to investigating cell degradation at low temperature and proposing approaches for performance enhancement[4,5]. Several mechanisms are generally responsible for poor cell performance at low temperatures: (a) a reduced rate of ion transfer in the electrode materials; (b) lower-than-usual electronic conductivity of the electrodes, especially for the widely used metal oxide electrodes; (c) lower-than-usual ionic conductivity of the electrolyte; and (d) sluggish charge-transfer kinetics induced by a decreased rate of chemical reactions. Fading electrochemical performance at low temperatures can be mitigated by introducing electrolyte additives[6], coating surfaces with some material that is highly electronically conductive[7,8], and heteroatom doping[9,10], but the attenuation is inevitable, and to date the highest retention of 86% at −25 °C for a sodium ion cell is achieved by Goodenough's group using an organic electrolyte[5]. Although this progress is valuable, for a cell required to operate over a long period in a cold climate, performance degradation remains unavoidable[5]. In addition, previously proposed methods of low-temperature cell performance enhancement largely focus on electrolyte modification, which cannot solve the sluggish kinetics of electrode reactions.

Recently, a family of materials, namely topological insulators with unique thermal and electrical properties, gain widespread interests[11,12]. These particular insulators have exotic metallic states formed by topological protection; their interiors function as insulators, but their surfaces act as conductors because their electrons are subject to strong spin-orbit interaction existing on their electrons[13]. Moreover, topological effects give these materials an inherent advantage: their properties are constant under any deformations. $Bi_2Se_3$, a topological insulator with a 0.3-eV nontrivial bulk gap and superficial single Dirac cone, is formed by periodic layers composed of five atomic planes (Se-Bi-Se-Bi-Se; namely, a quintuple layer denoted QL) connected through van der Waals forces; numerous unoccupied tetrahedral and octahedral exist between the Se atomic planes and result in the material's potential as an intercalating cathode in batteries[14,15]. Especially, for low-dimensional $Bi_2Se_3$, the reduced concentration of bulk carriers highlights the contribution of conductivity from surface topological states. Particularly, the conductivity of most of the $Bi_2Se_3$ nanoribbon devices increases as temperature goes down[16], and this metallic behavior provides an opportunity to circumvent the issue of cell electrodes operating at low temperatures. As topological surface states of $Bi_2Se_3$ protected by time inversion symmetry are robust at near ambient temperatures with exposure to liquids[17–22], topological quantum states of $Bi_2Se_3$ have potential applications in cell environments.

Herein, we report an aqueous Zn ion cell (AZIB) consisting of a few QLs of $Bi_2Se_3$ nanosheets (E-$Bi_2Se_3$) as the cathode, a Zn anode, and an antifreeze polyacrylamide (PAM) hydrogel electrolyte with highly concentrated salts incorporated along with ethylene glycol (denoted HC-EGPAM). Bulk $Bi_2Se_3$ powder (P-$Bi_2Se_3$) is readily exfoliated into E-$Bi_2Se_3$ through hydrothermal intercalation to enhance the coupling advantage in topological surface states. Surprisingly, the Zn||E-$Bi_2Se_3$ cell exhibits abnormally low-temperature performance (a capacity of up to 524 mAh g$^{-1}$ at −20 °C and 0.3 A g$^{-1}$) that is even better than its performance at 25 °C (a capacity of 327 mAh g$^{-1}$ at 0.3 A g$^{-1}$) and all previously reported measurements of low-temperature batteries. Even at −40 °C, the capacity retentions remain to 106 and 113% compared to the capacities at 25 °C. Specifically, per molecular E-$Bi_2Se_3$ holds up to four Zn$^{2+}$ on discharge at 25 °C and six Zn$^{2+}$ at −20 °C along with greatly increased unit cell

parameter ($c$). The unusual performance is attributed to the topological nature of E-$Bi_2Se_3$ improving kinetics of the cell reactions at lower temperatures. This is confirmed in this study by the observed higher electronic conductivity and good ion diffusion of the topological E-$Bi_2Se_3$ cathode at lower temperatures. Interestingly, after intercalation of Zn$^{2+}$ into the E-$Bi_2Se_3$ cathode, enhanced contribution of trivial metal states for Zn$_x$Bi$_2$Se$_3$ is revealed and proved theoretically and experimentally by conducting an electrical behavior test, density-functional theory (DFT) calculation, and ab initio molecular dynamics (MD) simulation. We believe that the developed topological insulator electrode and unusually better cell performance at lower temperatures will provide opportunities for producing batteries that will operate for long periods in cold climates.

## Results

**Characterization of P-$Bi_2Se_3$ and E-$Bi_2Se_3$.** A few QLs of E-$Bi_2Se_3$ has been testified the existence of coupling-enhanced topological surface states in contrast to the bulk P-$Bi_2Se_3$. The single-phase rhombohedral P-$Bi_2Se_3$ structure and $D_{3d}^5$ ($R\bar{3}m$) space group of the prepared samples are confirmed using X-ray powder diffraction (XRD; Fig. 1a). Hydrothermal intercalation is used to exfoliate the P-$Bi_2Se_3$ into E-$Bi_2Se_3$ (Supplementary Fig. 1). Although LiOH is involved during the exfoliation, analyses of Bi and Se ions by inductively coupled plasma atomic emission spectrometry (ICP-AES) and Li ion by ICP-mass spectrometry (ICP-MS) evidence the Li/Bi/Se atomic ratio for the E-$Bi_2Se_3$ sample is 0.00:1.00:1.48, respectively, indicating the absence of Li in the final product (Supplementary Table 1). For the E-$Bi_2Se_3$, exfoliation causes the number of stacking layers in one $Bi_2Se_3$ particle reduced, and the intensity of related host peaks (such as the (006) peak) is considerably lower than that in the P-$Bi_2Se_3$ spectrum[23]. E-$Bi_2Se_3$ lattice parameters of $a = 4.14(3)$ Å, $b = 4.14(3)$ Å, and $c = 28.67(4)$ Å are obtained through Rietveld refinement, and these parameters favorably match the experimental profile displayed in Fig. 1b. The refined atomic models are illustrated in Fig. 1c and Supplementary Fig. 2 where the framework consists of covalently bonded QLs coupled through van der Waals interactions in the [001] direction, accommodating the intercalant at high density and providing large tunnels for ion diffusion at interstitial sites or interlayer galleries in the van der Waals gap[24]. Rietveld refinement of the XRD pattern of P-$Bi_2Se_3$ are likewise shown in 3, and the refined structure parameters are listed in Supplementary Table 2. The transmission electron microscopy (TEM) analysis of P-$Bi_2Se_3$ presented in Supplementary Fig. 4a indicates irregular granules ranging in size from 200 nm to 1 μm. The corresponding selected area electron diffraction (SAED) spot pattern indicates that the P-$Bi_2Se_3$ is polycrystalline (Supplementary Fig. 4b). After exfoliation, ultrathin and stacked nanosheets with ripples of E-$Bi_2Se_3$ and sized within 1 μm are obtained (Fig. 1d), and a typical hexagonal SAED pattern of E-$Bi_2Se_3$ consistent with the (110) and (300) planes is distinctly observed, confirming the high level of single crystallinity. The high-angle annular dark-field scanning transmission electron microscopy (HAADF-STEM) image of E-$Bi_2Se_3$ recorded along the [010] zone axis (Fig. 1e) is a magnified view from the A region of Supplementary Fig. 5a, where 1 QL is well assigned to the layered structure. The corresponding electron energy loss spectroscopy (EELS) elemental maps confirm the uniform distribution of Se and Bi (Fig. 1e). Magnified view of a rectangular area in Fig. 1e shows consistency with the E-$Bi_2Se_3$ rhombohedral structure at the atomic resolution (Supplementary Fig. 5b–d and Fig. 1f)[25,26]. A structural model is shown on the left of the HAADF image. Figure 1g shows the intensity line scan along the orange dashed line in the HAADF image. Clear peaks

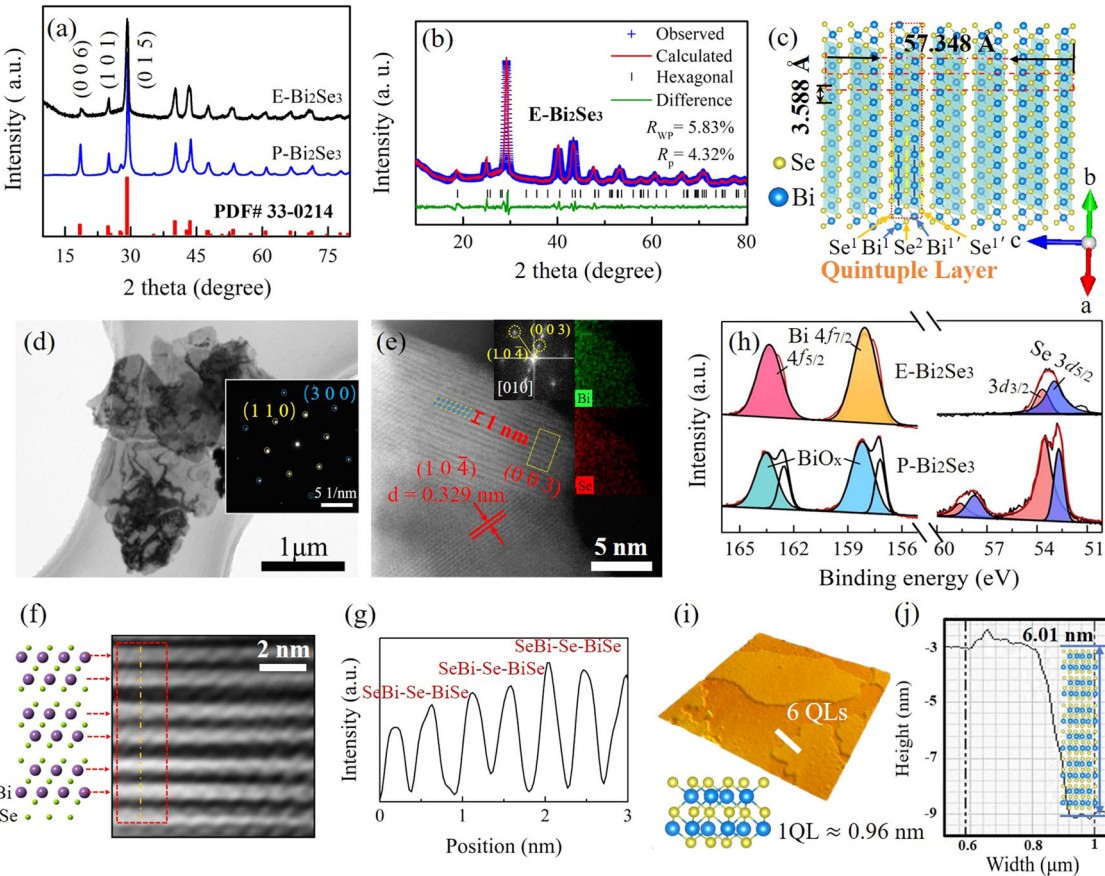

**Fig. 1 Structural and morphological characterization of prepared P-Bi₂Se₃ and E-Bi₂Se₃. a** XRD analyses of P-Bi₂Se₃ and E-Bi₂Se₃ (matched with powder diffraction file cards No. 00-033-0214). **b** Rietveld refinement of the XRD pattern of E-Bi₂Se₃ with reliability factors $R_{wp}$ and $R_p$ of 5.83% and 4.32%, respectively. **c** Layered crystal structure of E-Bi₂Se₃ with quintuple layers aligned perpendicular to the trigonal c-axis. The dashed red square indicates a quintuple layer in the Se¹-Bi¹-Se²-Bi¹'-Se¹' sequence, and the dimensions shown here are calculated from the lattice constants. **d** TEM image of E-Bi₂Se₃ and corresponding SAED pattern. **e** HAADF- STEM image of E-Bi₂Se₃ and its corresponding EELS elemental mappings along [010] zone axis. **f** A magnified view of the yellow rectangular area in **e** with the E-Bi₂Se₃ crystal structure. **g** An intensity profile from orange line in **f**. **h** XPS spectra of Bi 4f and Se 3d signals for P-Bi₂Se₃ and E-Bi₂Se₃. **i** Typical AFM image of individual E-Bi₂Se₃ and **j** corresponding height profile.

are observed due to the Se-Bi-Se atoms column. We averaged several plots across Se-Bi-Se cluster along pink dashed lines (Supplementary Fig. 5e). Assume the atom is a Gaussian shape. The averaged plot (average of 8 line profiles) is fitted with Gaussians (Supplementary Fig. 5f). Two Se atoms do not distribute symmetrically around Bi site. The phenomena are due to that the atomic configuration at different areas may have certain distortion after HEMM and hydrothermal exfoliating processes as capturing the atomic-scaled HAADF image along [010] zone axis (see Supplementary Fig. 5 for details). Two peaks together in Fig. 1g corresponds to a QL of Se-Bi-Se-Bi-Se with a thickness approaching 1 nm, confirming the ordered arrangement of QLs originating from the layered structure. High-resolution HAADF-STEM image from the B region of Supplementary Fig. 5a shown in Supplementary Fig. 5g has a (003) crystal plane orientation confirmed by corresponding Fourier transform (FT) pattern, which highlights the lamellar structure of E-Bi₂Se₃ (Supplementary Fig. 5h). The electron energy loss spectrum of E-Bi₂Se₃ shows that E-Bi₂Se₃ does not contain Li as there is no peak at 55 eV in Supplementary Fig. 5i. X-ray photoelectron spectroscopy (XPS) also indicate the composition of P-Bi₂Se₃ and E-Bi₂Se₃ (Supplementary Fig. 6a). No liquid intercalation was observed (Supplementary Fig. 6b). The high-resolution spectra reveal the valence states of the Bi and Se (Fig. 1h). The Bi 4f spectrum of E-Bi₂Se₃ has two peaks at 163.37 and 158.02 eV, corresponding to the Bi 4f₅/₂ and 4f₇/₂ binding energies attributable to Bi₂Se₃, respectively,

whereas deconvolution of the Bi 4f core-level peaks in the P-Bi₂Se₃ spectrum indicates that the Bi 4f₅/₂ and 4f₇/₂ peaks are shifted to higher energies by 0.9 eV, which is likely attributable to the formation of oxides (BiOₓ), as discovered in a previous study on Bi₂Se₃ nanowires signifying higher oxidation states than those of E-Bi₂Te₃[27]. Moreover, one peak of Se 3d level for E-Bi₂Se₃ reveal the contributions of Se²⁻ in Bi₂Se₃ from the Se 3d₅/₂ level at binding energy of 52.85 eV, and Se 3d₃/₂ level at 53.74 eV, and a similar phenomenon emerges in P-Bi₂Se₃. The blueshift of the Se 3d peak at 58.2 eV in P-Bi₂Se₃ is attributed to amorphous SeO₂ introduced during HEMM process[28,29] (Supplementary Fig. 7). After washing with tetramethylammonium hydroxide (TMAH)/ NaOH/NaCl aqueous solution and DI water, SeO₂ impurity can be fully removed. Figure 1i presents a typical atomic force microscopy (AFM) image showing the smooth surface of E-Bi₂Se₃, and Fig. 1j shows the corresponding height profile, revealing that the thickness of E-Bi₂Se₃ is approximately 6 nm and that the material thus consists of 6 QLs given than the thickness of a QL is approximately 0.96 nm[30]. The thickness of numerous flakes is calculated from the AFM image in Supplementary Fig. 8, the average thickness is statistically estimated to be 6 nm.

**Electrochemical performance comparison in Zn‖P-Bi₂Se₃ and Zn‖E-Bi₂Se₃ cells.** The temperature-dependent electrochemical

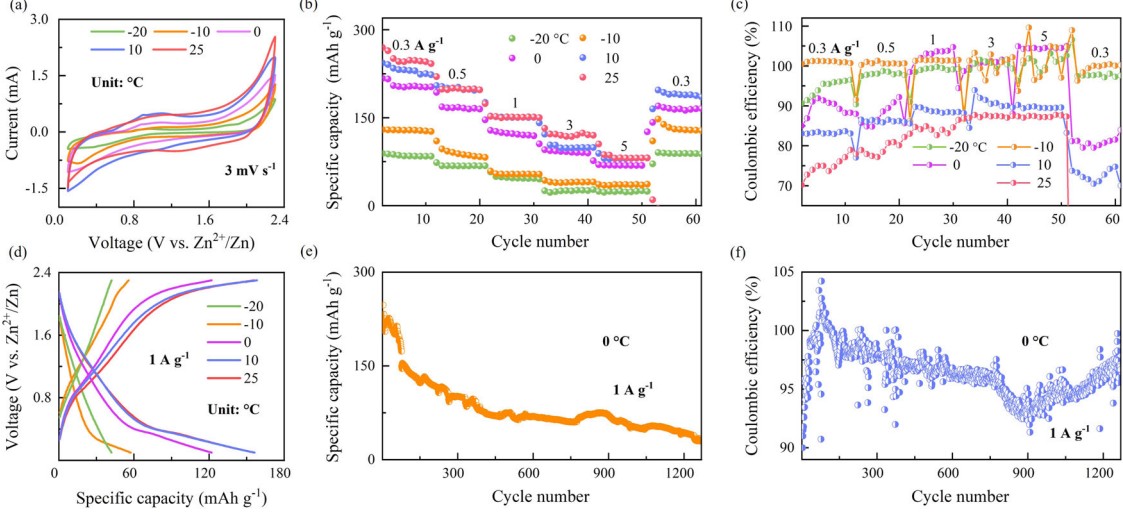

**Fig. 2 Electrochemical performance of the rechargeable quasi-solid Zn||P-Bi$_2$Se$_3$ cells over the temperature range −20 to 25 °C. a** CVs for quasi-solid Zn||P-Bi$_2$Se$_3$ cells at different temperatures. **b** Rate capability and corresponding CE (**c**) and GCD profiles (**d**) of the quasi-solid AZIB at different temperatures. **e** Long-cycle performance and corresponding CE (**f**) at 0 °C.

performance of the rechargeable Zn||P-Bi$_2$Se$_3$ and Zn||E-Bi$_2$Se$_3$ cells is characterized using HC-EGPAM as the electrolyte. The preparation and characterization of HC-EGPAM hydrogel electrolyte are detailed in Supplementary Fig. 9. The HC-EGPAM hydrogel has high adhesiveness, favorable freezing tolerance, and flexibility even at −35 °C (Supplementary Fig. 9h–j)[6]. The ionic conductivity of HC-EGPAM hydrogel is much higher than that of PAM containing highly concentrated salts (HC-PAM) at low temperatures (Supplementary Fig. 9f). However, when the temperature is decreased from 25 to −20 °C, 50% of the ionic conductivity of HC-EGPAM is still lost (from 7.85 to 3.90 ms cm$^{-1}$).

We first test the electrochemical performance of the quasi-solid Zn||P-Bi$_2$Se$_3$ cell, in which the P-Bi$_2$Se$_3$ exhibits limited topological insulating states. Figure 2a shows a typical cyclic voltammogram (CV) with weakened intensity at decreased temperature. The temperature-dependent rate performance is illustrated in Fig. 2b. At 25 °C, capacities values of 249, 199, 152, 129, and 88 mAh g$^{-1}$ are obtained at current rates of 0.3, 0.5, 1, 3, and 5 A g$^{-1}$, respectively. At lower temperature, the reversible capacity of the Zn||P-Bi$_2$Se$_3$ cell is severely decreased. At 0.3 A g$^{-1}$, the Zn||P-Bi$_2$Se$_3$ cell maintains reversible capacities values of 231, 205, 130, and 87 mAh g$^{-1}$ at 10, 0, −10, and −20 °C, respectively, which are 92%, 82%, 52%, and 35% of the capacities delivered at 25 °C, respectively. At higher temperature or lower specific current, characteristic low coulombic efficiency (CE) is noted on account of the irreversible side reaction (Fig. 2c)[31]. Figure 2d displays the corresponding galvanostatic charge/discharge (GCD) curves obtained at various temperatures from −20 to 25 °C at 1 A g$^{-1}$. Discharge capacities of 42, 57, 122, 156, and 156.3 mAh g$^{-1}$ are obtained at temperatures of −20, −10, 0, 10, and 25 °C, respectively. The GCD profile obtained at −20 °C is almost a straight line, which is attributable to the sluggish Zn$^{2+}$-transport kinetics[32]. The Zn||P-Bi$_2$Se$_3$ cell exhibits poor cycling performance at a low temperature of 0 °C in Fig. 2e, f: the capacity retention is less than 16% after 1200 cycles at 1 A g$^{-1}$, in which P-Bi$_2$Se$_3$ electrode delamination leads to the cell failure[33]. Our observation clearly indicates that even when an anti-freeze electrolyte is employed, performance degradation of Zn||P-Bi$_2$Se$_3$ at low temperature is unavoidable, which is similar to other reported cells. The working mechanism of P-Bi$_2$Se$_3$ is studied in Supplementary Fig. 10, and the capacitive behavior with poor bulk diffusion is observed.

We further evaluate the electrochemical performance of the as-prepared E-Bi$_2$Se$_3$ cathodes in an AZIB at different temperatures with the E-Bi$_2$Se$_3$ exhibiting enhanced topological states[16,34]. Unlike what we have observed for the Zn||P-Bi$_2$Se$_3$ cell, as the temperature is decreasing, the cathodic peaks shift to higher potential and anodic peaks shift to lower potential, which can be ascribed to reduced polarization (Fig. 3a)[35]. Exceptionally, the area of redox peaks is remarkably enlarged at lower temperature, indicating higher reaction kinetics of E-Bi$_2$Se$_3$ electrodes at low temperature than at 25 °C[36]. Figure 3b, c depict the good rate capability of Zn||E-Bi$_2$Se$_3$ cells at −20 to 25 °C. Unlike all previously reported cells, including the Zn||P-Bi$_2$Se$_3$ cells for which attenuation inevitably occurs at low temperatures, the Zn||E-Bi$_2$Se$_3$ cells anomaly deliver better performances at lower temperatures. At −20 °C, discharge specific capacities of 526.3, 400.9, 301.3, and 206.1 mAh g$^{-1}$ are obtained at specific currents of 0.3, 0.5, 1.0, and 3.0 A g$^{-1}$, respectively. Compared with the capacities of 326.7, 278, 231, and 159 mAh g$^{-1}$ at 25 °C, the capacity retention values at −20 °C are 161%, 144%, 130%, and 129.6%, respectively. When the current rate returns to 0.3 A g$^{-1}$ from 3 A g$^{-1}$, discharge capacities of 499 (the rate capacity retention of 95%), 468, 405, 335, and 317 mAh g$^{-1}$ are recovered at −20, −10, 0, 10, and 25 °C, respectively. The corresponding GCD profiles at different temperatures at 1 A g$^{-1}$ are displayed in Fig. 3d, all containing a discharge plateau at approximate 1.6 V and a subsequent slope at around 1.0 V. In sharp contrast to the Zn||P-Bi$_2$Se$_3$ cells, the Zn||E-Bi$_2$Se$_3$ cells exhibit surprisingly enhanced discharge capacity (269 mAh g$^{-1}$) and cycling capacity retention (94.6% after 2000 cycles at 0 °C) at 1 A g$^{-1}$ with a high CE nearly 100% (Fig. 3e, f). The capacity in Fig. 3e sharply increases in the first tens cycles due to the activation of the electrode. In addition, the polarization of the cell is negligible, being as low as 0.08 V at −20 °C. We further extend the temperature range (Supplementary Fig. 11). At the temperature ranging from −20 °C to −50 °C (or from 50 °C to 30 °C), electrolyte has a greater influence on the cell performance (Supplementary Fig. 11a–d) as the ionic conductivity may remarkably decrease with the declined temperature[37]. For the region from 30 °C to −20 °C, the contribution of E-Bi$_2$Se$_3$ electrode to the cell performance enhancement is greater than that of the electrolyte (Supplementary Fig. 11e). Supplementary Figure 11f summarizes capacity retention of Zn||E-Bi$_2$Se$_3$ cells at

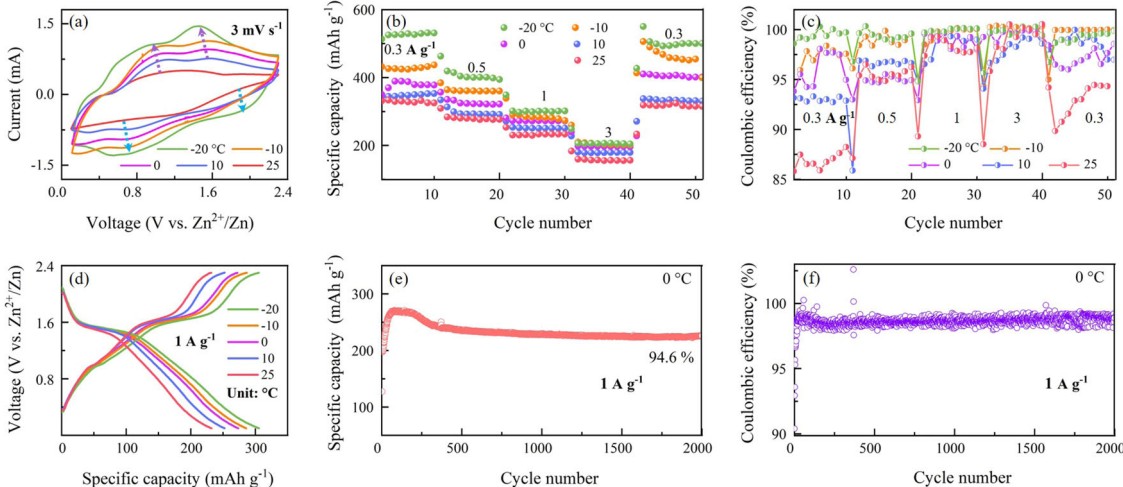

**Fig. 3 Electrochemical performance of the rechargeable quasi-solid Zn||E-Bi$_2$Se$_3$ cells over the temperature range −20 to 25 °C. a** CVs of quasi-solid Zn||E-Bi$_2$Se$_3$ cells at different temperatures. **b** Rate capability and corresponding CE (**c**) and GCD profiles (**d**) of the quasi-solid AZIB at different temperatures. **e** Long-cycle discharge capacity and corresponding CE (**f**) at 0 °C.

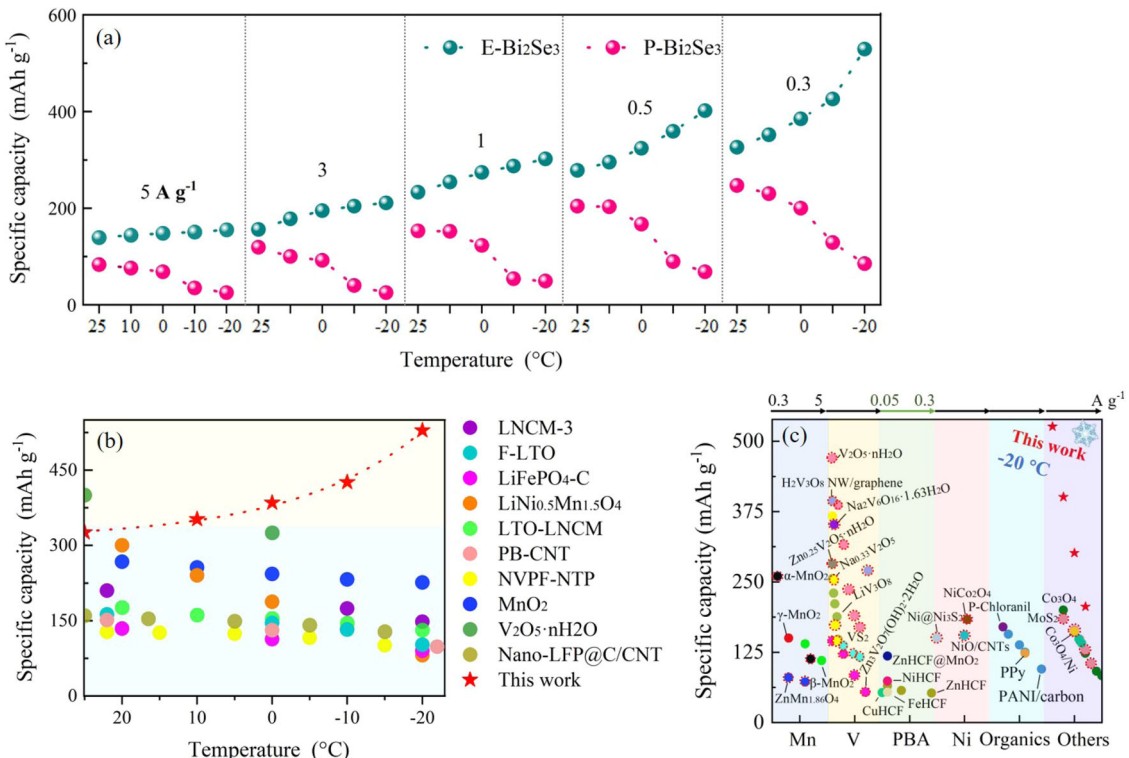

**Fig. 4 Comparison of specific capacity of this work with that of previous reports. a** Comparison of discharge specific capacity versus temperature of Zn|| P-Bi$_2$Se$_3$ and Zn||E-Bi$_2$Se$_3$ cells at various current rates and temperatures. **b** Specific capacity versus temperature of various low-temperature cells including lithium/sodium/zinc ion cells, among which LNCM-3, F-LTO, LTO-LNCM, PB-CNT, NVPF-NTP, and LFP stand for LiNi$_{0.6}$Co$_{0.2}$Mn$_{0.2}$O$_2$[7], fluorinated Li$_4$Ti$_5$O$_{12}$[9], Li$_4$Ti$_5$O$_{12}$/Li[Ni$_{0.45}$Co$_{0.1}$Mn$_{1.45}$]O$_4$[3], Prussian blue-carbon nanotubes[5], Na$_3$V$_2$(PO$_4$)$_2$O$_2$F nanotetraprisms[32], and LiFePO$_4$[39], respectively. **c** Comparison of specific capacity between the present E-Bi$_2$Se$_3$ cathode at −20 °C and representative cathodes in AZIBs at room temperature and various current rates. All the current rates herein are in the range 0.3–5.0 A g$^{-1}$ except for Prussian blue analogs (PBAs), for which they are 0.05–3.00 A g$^{-1}$. The exfoliated and nanoscale electrodes are circled with dotted lines.

various temperatures and current rates compared to the corresponding discharge capacities at 25 °C. Remarkably, even at −40 °C the capacity retentions at 1 and 3 A g$^{-1}$ remain to be over 100% (106 and 113 %, respectively). The Zn||E-Bi$_2$Se$_3$ cells still deliver discharge capacities of 280, 245, 206, 140, and 113 mAh g$^{-1}$ at specific currents from 0.3 to 5 A g$^{-1}$ at −50 °C

with record-high capacity retentions of 86, 88, 88, 90, and 81%, respectively (Supplementary Fig. 11f).

Figure 4a and Supplementary Fig. 12a summarize comparisons of temperature-dependent discharge gravimetric and volumetric capacities of Zn||E-Bi$_2$Se$_3$ and Zn||P-Bi$_2$Se$_3$ cells at various current rates, respectively. Clearly, unlike the Zn||P-Bi$_2$Se$_3$ with

a degraded performance at lower temperatures, the specific capacity of Zn||E-Bi$_2$Se$_3$ increases as the temperature decreases. To rule out the contribution from the electrolyte, we compare Zn||MnO$_2$ cells using HC-EGPAM with other reported antifreeze hydrogel (Supplementary Fig. 12b), and also compare our topological insulator E-Bi$_2$Se$_3$ electrode with PBAs and MnO$_2$ electrodes in other antifreeze hydrogel electrolytes (Supplementary Fig. 12c, details of discussion are given in the Supporting Information). This anomalous low-temperature cell performance has also been observed in a Na$_3$VCr(PO$_4$)$_3$ (NVCP) cathode[38]. However, it should be noted that this is because NVCP is not stable at a relatively high temperature (30 °C), leading to a poorer performance output at higher temperature. At low temperature, the high temperature induced irreversible electrochemical reaction can be suppressed in NVCP[38]. With a totally different mechanism, here exfoliation endows Bi$_2$Se$_3$ with large electrode/electrolyte interfacial contact areas (reduced the ion diffusion length) and fast ion diffusion, which improves the cells performance. More importantly, exfoliation enhances the topological protection performance of the electrodes, significantly increasing electronic conductivity and boosting the electron transfer kinetics (Supplementary Fig. 13)[13,16]. What really gives the cell remarkably improved performance at low temperature is the synergistic effect of exfoliation and exfoliation-enhanced topological surface state. Compared with all other low-temperature cells including lithium/sodium/zinc ion cells in aqueous[6,39] or organic[3,5,7,9,32,40,41] electrolytes, as shown in Fig. 4b, the temperature-dependent capacity of Zn||E-Bi$_2$Se$_3$ is among the best. Moreover, the variation trend of capacity versus temperature is clearly reversed. Figure 4c discloses the comparison profile of the dependence of specific capacity on current rate for the prepared E-Bi$_2$Se$_3$ cathode and representative Mn[42,43], V[44–48], and Ni-based cathodes[49,50], PBAs[51,52], organic cathodes[35,53,54], and others[55,56] in AZIBs. The Zn||E-Bi$_2$Se$_3$ possesses the highest specific capacity of 526.3 mAh g$^{-1}$ at 0.3 A g$^{-1}$ at −20 °C, which is even prominently outstrip those of the most room-temperature reports. The E-Bi$_2$Se$_3$ cathode is also amenable to high mass loadings of 1–4 mg cm$^{-2}$. They are tested at specific currents of 0.1 A g$^{-1}$ to 5 A g$^{-1}$ at different temperatures. All samples exhibit a higher capacity at lower temperatures, confirming our previous observation (see Supplementary Fig. 14a–d for details). A high areal capacity of 2.1 mAh cm$^{-2}$ (0.1 A g$^{-1}$) at a mass loading of 4 mg cm$^{-2}$ and −20 °C is achieved. Besides, we also assemble flexible quasi-solid Zn||E-Bi$_2$Se$_3$ cells with electrodeposited Zn on carbon cloth as the anode (Supplementary Fig. 15), which can successfully power a wristwatch at low temperature. Considering high-voltage applications, we boost the Zn||E-Bi$_2$Se$_3$ cell voltage to 3.3 V through a DC-DC boost converter with a transition efficiency of 90% (Supplementary Fig. 16a). We first charge Zn||E-Bi$_2$Se$_3$ to 2.3 V, then connect it to the converter following by discharging at −20 °C. Even at 3.3 V, considerable capacity of 438 mAh cm$^{-3}$ (0.5 A g$^{-1}$), and 386 mAh cm$^{-3}$ (1 A g$^{-1}$) are obtained, verifying the low-temperature application potential. Zn||E-Bi$_2$Se$_3$ exhibits a maximum specific energy of 441 Wh kg$^{-1}$ at a specific power of 683 W kg$^{-1}$, which is remarkable better than that of other reported aqueous AZIBs, especially at low temperature (Supplementary Fig. 16b).

## The Zn-ion storage mechanistic study of E-Bi$_2$Se$_3$.
Our findings clearly indicate that topological insulating states play a crucial role in the unique low-temperature performance of the Zn||E-Bi$_2$Se$_3$ system. To reveal the mechanism underlying this performance, we first employed ex situ XRD to investigate the E-Bi$_2$Se$_3$ cathode during charge and discharge. Figure 5a presents the XRD

patterns of the E-Bi$_2$Se$_3$ cathode at selected charge/discharge states corresponding to the GCD profile at 0.3 A g$^{-1}$ in the liquid electrolyte (see Supplementary Fig. 17 for electrochemical performance of the rechargeable Zn||E-Bi$_2$Se$_3$ cells in 1 m Zn(TFSI)$_2$ and 21 m LiTFSI/H$_2$O aqueous electrolyte), and magnified XRD patterns of characteristic peaks are shown in Fig. 5b. During the discharge process (samples 1–8 in the red region), three sets of peaks at 17.7°, 24.7°, and 43.4° [indexed to the (006), (101), and (110) planes of Bi$_2$Se$_3$] shift to lower angles, indicating that the intercalation of Zn$^{2+}$ triggers expansion of the interlayer spacing. During subsequent charging of the cell to 2.3 V (samples 9–16 in the blue region), these peaks gradually migrate back to their initial positions, reflecting the highly reversible crystal structure evolution of E-Bi$_2$Se$_3$. The peaks at approximately 29°, 40°, and 53.2°— assigned to the (015), (1 0 10), and (205) planes — remains almost unchanged during the charge-discharge process (Supplementary Fig. 18). Ex situ Raman spectra are obtained to analyze the cathode's structural stability during the electrochemical process (Supplementary Fig. 19). The representative Raman-active modes of Bi$_2$Se$_3$ ($^1A_{1g}$, $^2E_g$, and $^2A_{1g}$ located at 72, 132, and 174 cm$^{-1}$, respectively) are enlarged in Fig. 5c. The corresponding measurements of intensity and full width at half maximum (FWHM) of Raman-active modes ($^1A_{1g}$, $^2E_g$, and $^2A_{1g}$) in Supplementary Fig. 19c reveals that upon discharge to 0.1 V, the three modes become broader and weaker, which corresponds to the embedding of Zn$^{2+}$ into the interlayers of E-Bi$_2$Se$_3$. When charging to 2.3 V, these characteristic peaks gradually become stronger and finally return to their initial states, which can be attributed to the deintercalation of Zn$^{2+}$ from the electrode framework. The Raman results further rule out solvent insertion, confirming that Zn$^{2+}$ is exclusively responsible for the enlarged spacing. TEM is conducted to probe the structural and morphological evolution of E-Bi$_2$Se$_3$ after it has been fully discharged, presenting the interlaced ultra-thin nanosheets (Fig. 5d). Under higher magnification in Fig. 5e with SAED patterns in Supplementary Fig. 20a, b, the high crystallinity of E-Bi$_2$Se$_3$ is accompanied by an increase in the interplanar spacing from 0.206 to 0.215 nm, indexed to the (110) plane. Also, a characteristic superlattice pattern is observed in which a hexagon of six superlattice spots surrounds each of the host lattice spots as shown in Fig. 5f and Supplementary Fig. 20c. The experimental intensity of the spots and the interplanar spacing are analyzed to be 0.274 and 0.160 nm which corresponds to the (107) and (0 2 10) planes of Zn$_x$Bi$_2$Se$_3$, respectively. These planes are in good agreement with those of the simulated electron diffraction pattern of Zn$_4$Bi$_2$Se$_3$ crystal taken along the same zone axis of [001] demonstrating the intercalated Zn$^{2+}$ in the interlayer (Supplementary Fig. 20d). While there are no significant changes in the spacing of (018), (015), and (107) planes (Fig. 5g and Supplementary Fig. 21), which is in agreement with the XRD results. The HAADF-STEM image taken from a red dotted rectangular box in Supplementary Fig. 22a is shown in Fig. 5h. The layered structure of Zn$_x$Bi$_2$Se$_3$ follows the layered Zn$_x$Bi$_2$Se$_3$ lattice (Supplementary Fig. 22b). Inset is an FFT pattern showing the (003) crystallographic plane. Figure 5i is a magnified view from a rectangular area from Fig. 5h. Bi atom is located in the middle position between two Se atoms (Supplementary Fig. 22c–f). Figure 5j shows an intensity profile from the magnified part of Fig. 5i. The Zn$^{2+}$ occupies the position between the Se layers (quantification of the HAADF image is discussed in detail in Supplementary Fig. 22e, f. Comparison of the vertical line scan profiles of the HAADF images of E-Bi$_2$Se$_3$ and Zn$_x$Bi$_2$Se$_3$ is shown in Supplementary Fig. 22g. TEM-EDS elemental mappings (Fig. 5k) and corresponding TEM-EDS spectrum (Supplementary Fig. 23) reveal that Zn uniformly intercalates in the E-Bi$_2$Se$_3$ nanosheets. The electrode remains in

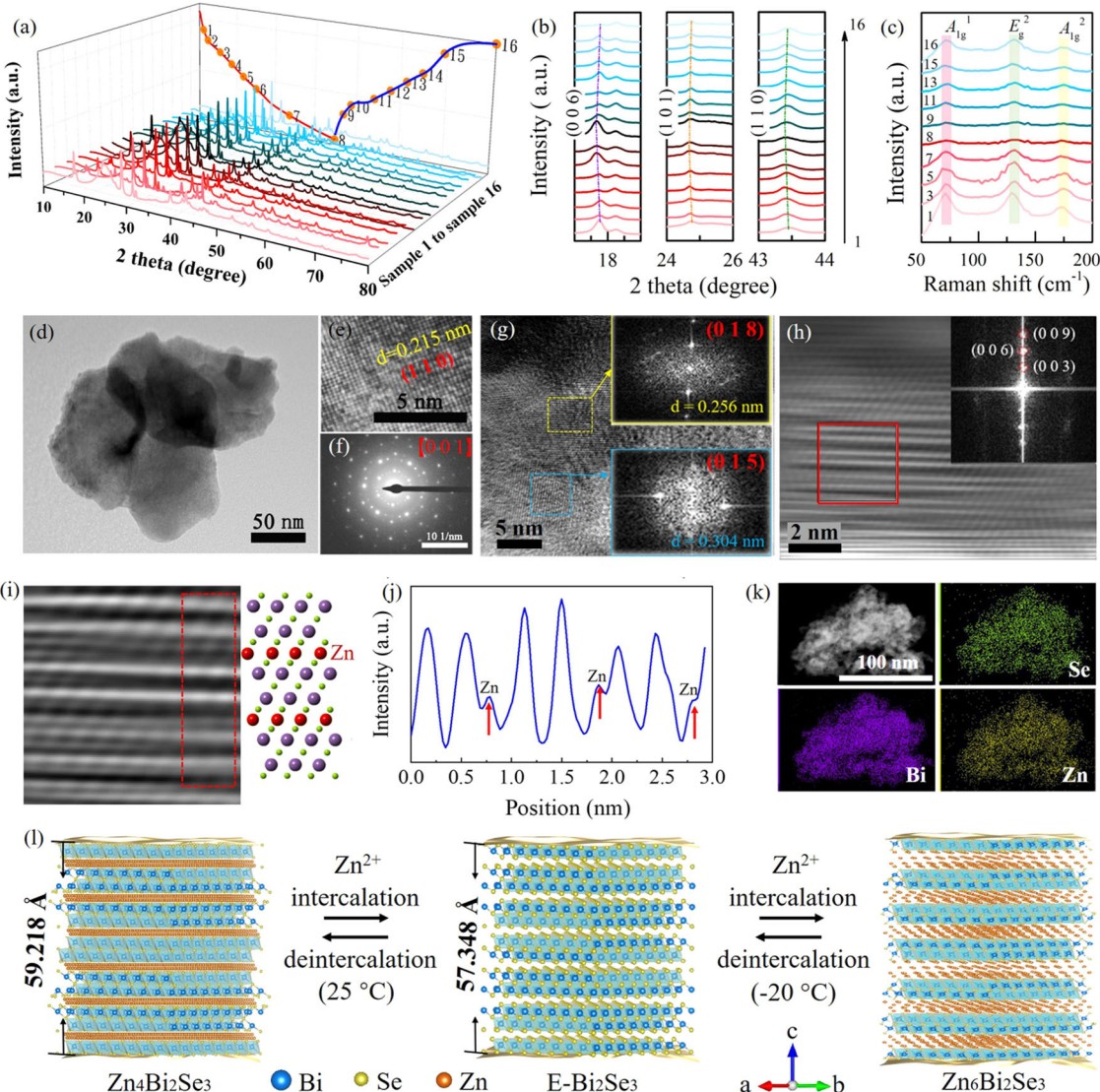

**Fig. 5 Structural evolution of the E-Bi$_2$Se$_3$ cathode during electrochemical cycling. a** Evolution of ex situ XRD patterns during the charge/discharge process for E-Bi$_2$Se$_3$ for the third cycle in the liquid electrolyte at 0.3 A g$^{-1}$ (the solid dots numbered sequentially from 1 to 16 refer to the positions at which the XRD patterns are obtained, and the red region is corresponding to the discharge process while the blue corresponding to the charge process). **b** Magnified XRD patterns from **a** for 17°–19°, 24°–26°, and 43°–44°. **c** Magnified part (50–200 cm$^{-1}$) of ex situ Raman spectra of the E-Bi$_2$Se$_3$ at the selected states corresponding to the charge/discharge process illustrated in **a**. **d** TEM image and **e** high-resolution TEM (HRTEM) image of E-Bi$_2$Se$_3$ at fully discharged state after three cycles. **f** SAED pattern of E-Bi$_2$Se$_3$ at fully discharged state. **g** HRTEM image of fully discharged E-Bi$_2$Se$_3$ revealing (018) and (015) planes with insets displaying fast Fourier transform patterns. **h** HAADF-STEM image of Zn$_x$Bi$_2$Se$_3$, and its corresponding FT pattern. **i** Magnified part from the red box in **h** with the Zn$_x$Bi$_2$Se$_3$ crystal structure and **j** corresponding intensity profile. The red arrows in **j** indicate the presence and specific location of Zn$^{2+}$ sites. **k** TEM-EDS elemental maps of Bi, Se and Zn are given in colors of purple, green and yellow dots, respectively. **l** Schematic of Zn$^{2+}$ intercalation and deintercalation in the E-Bi$_2$Se$_3$ cathode upon electrochemical charge and discharge process at −20 or 25 °C. Gold sequins represent topological surface states.

the original pattern of ultrathin interconnected nanosheets anchored stably on the base (see Supplementary Fig. 24 for different magnifications), and Supplementary Fig. 25 depicts the Zn 2$p$ core-level spectra of the E-Bi$_2$Se$_3$ cathode in three stages. In its original state, the spectrum doesn't manifest the Zn 2$p_{1/2}$-2$p_{3/2}$ spin-orbit doublet, whereas the spectrum of the cathode discharged to 0.1 V confirms successful intercalation of Zn$^{2+}$ (insertion state), and the weak signal in the spectrum obtained for the cathode on charge clearly reveals the retention of a few intercalated Zn$^{2+}$ (extraction state), which offers clear evidence of reversible Zn$^{2+}$ intercalation/deintercalation into the E-Bi$_2$Se$_3$ cathode.

To gain additional insights into the potential mechanism underlying the unique low-temperature electrochemical properties,

we investigate the Zn$^{2+}$ insertion model, electronic structures, and Zn$^{2+}$ ion diffusion kinetics for E-Bi$_2$Se$_3$ by using DFT calculations and ab initio MD simulation. A reversible crystal structure transformation between E-Bi$_2$Se$_3$ and Zn$_x$Bi$_2$Se$_3$ during the charge-discharge process is illustrated in Fig. 5l. During the discharge process, the discharge capacity of 327 mAh g$^{-1}$ under a specific current of 0.3 A g$^{-1}$ at 25 °C corresponds to the Zn insertion phase, with the stoichiometry of Zn$_4$Bi$_2$Se$_3$ amounting to almost 99.88% of the calculated capacity (327.4 mAh g$^{-1}$), and strikingly, the low-temperature (−20 °C) discharge capacity of 524 mAh g$^{-1}$ at 0.3 A g$^{-1}$ corresponds to Zn$_6$Bi$_2$Se$_3$. After intercalation at 25 °C, the unit cell parameter ($c$, perpendicular to QLs) increases from 29.16 to 32.92 Å (Zn$_4$Bi$_2$Se$_3$). Upon the

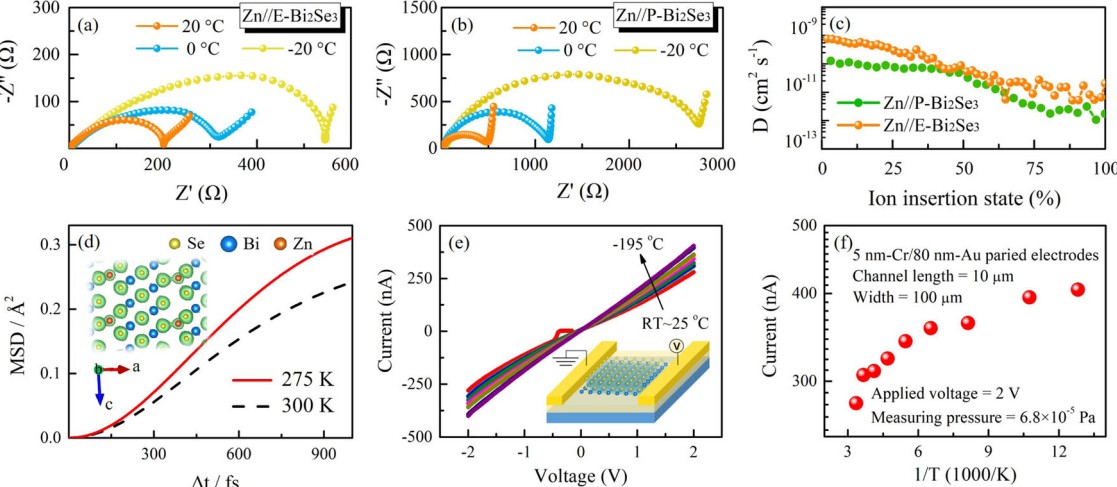

**Fig. 6 Study of the low-temperature electronic conductivity and ion diffusion kinetics of E-Bi$_2$Se$_3$ cathode.** EIS of the **a** Zn||E-Bi$_2$Se$_3$ and **b** Zn||P-Bi$_2$Se$_3$ cells, obtained at −20, 0, and 20 °C. **c** Diffusivity coefficient (D) of Zn$^{2+}$, calculated using the GITT, of Zn||E-Bi$_2$Se$_3$ and Zn||P-Bi$_2$Se$_3$ cells in the third cycle at 25 °C. **d** MSD of the Zn between 4 × 4 × 1 supercell Bi$_2$Se$_3$ bilayers, as discovered using ab initio MD simulation at 275 and 300 K, respectively. The inset displays electron density plot for the Zn$_x$Bi$_2$Se$_3$ skeleton. **e** I–V curves of Zn$_x$Bi$_2$Se$_3$ with inset showing schematic of the Zn$_x$Bi$_2$Se$_3$ nanosheet device fabricated for the electrical transport experiment and **f** the corresponding temperature-dependent current analysis on the Zn$_x$Bi$_2$Se$_3$ device.

subsequent charge process, Zn$^{2+}$ is reversibly deintercalated from Zn$_4$Bi$_2$Se$_3$ and, together with two electrons, is transformed into the original Zn. Overall, Supplementary Fig. 26 depicts a schematic of the full rechargeable Zn∣∣E-Bi$_2$Se$_3$ cell.

**Low-temperature electronic conductivity and ion diffusion kinetics study of E-Bi$_2$Se$_3$.** Because the performance of the electrolyte is worse at low temperatures than that at room temperature, it is believed that the enhanced electrochemical performances of Zn||E-Bi$_2$Se$_3$ cells is mainly contributed by improved electrode properties. We then investigate the electronic conductivity and ion diffusion kinetics of E-Bi$_2$Se$_3$ electrodes at different temperatures. The electrochemical impedance spectra (EIS) of Zn||E-Bi$_2$Se$_3$ (Fig. 6a) and Zn||P-Bi$_2$Se$_3$ cells (Fig. 6b) are conducted, and the fitting results with corresponding equivalent circuits are summarized in Supplementary Fig. 27. The $R_i$ of Zn|| E-Bi$_2$Se$_3$ is much smaller relative to that of Zn||P-Bi$_2$Se$_3$, especially at −20 °C (252 vs. 2320 Ω), indicating the far better wettability of the E-Bi$_2$Se$_3$ electrode for the HC-EGPAM electrolyte, caused by the enlarged interlayer spacing. The slightly increased $R_s$ of Zn||E-Bi$_2$Se$_3$ from 6.2 Ω at 20 °C to 8.55 Ω at −20 °C—which involves the resistance of the HC-EGPAM, separator, and E-Bi$_2$Se$_3$ electrode—reflects the good electric conductivity of the cells. The faradic impedance (namely the combination of $R_{ct}$ and $Z_W$) reflects the kinetics of the cell reactions[57]. Herein, the considerably lower $R_{ct}$ at −20 °C (288 Ω) of Zn||E-Bi$_2$Se$_3$ than the $R_{ct}$ of Zn||P-Bi$_2$Se$_3$ (500 Ω) favorably escorts fast Zn$^{2+}$ diffusion kinetics. Additionally, $Z_w$ is assigned to the Zn$^{2+}$ diffusion in the cell[58]. As the slope is proportional to $Z_w$, the Zn||E-Bi$_2$Se$_3$ cell with smaller slopes incidates a faster Zn$^{2+}$ diffusion[59]. These results reveal that the stable ionic transport and higher electric conductivity of the Zn||E-Bi$_2$Se$_3$ cell in cold environments far surpass those of the Zn||P-Bi$_2$Se$_3$ cell.

In addition, the average Zn$^{2+}$ diffusion coefficient of the Zn||E-Bi$_2$Se$_3$ cell is calculated using the galvanostatic intermittence titration technique (GITT) to be 10$^{-10}$–10$^{-11}$ cm$^2$ s$^{-1}$, higher than that of the Zn∣∣P-Bi$_2$Se$_3$ cell (Fig. 6c and Supplementary Fig. 28). To unravel the low-temperature Zn$^{2+}$ diffusion of the E-Bi$_2$Se$_3$ cathode, we calculate the mean square displacement (MSD) by selecting last 4000-steps in the simulation. Figure 6d graphs the calculated MSD of the Zn ions between the bilayer

E-Bi$_2$Se$_3$ at 275 and 300 K. The MSD of Zn in E-Bi$_2$Se$_3$ at 275 K is clearly higher than that at 300 K. On the basis of the calculated MSD, we compute the diffusion coefficient of Zn ions within the bilayer Bi$_2$Se$_3$ at 275 and 300 K, 5.993 × 10$^{-5}$ m$^2$ s$^{-1}$ and 4.603 × 10$^{-5}$ m$^2$ s$^{-1}$, respectively, indicating that the Zn ions diffuse more quickly at the lower temperature (275 K) than the higher temperature (300 K) in the topological E-Bi$_2$Se$_3$. This phenomenon can be attributed to the weaker lattice vibration of the E-Bi$_2$Se$_3$ bilayer at 275 K than at 300 K, which facilitates the movement of Zn$^{2+}$. Additionally, the inset of Fig. 6d shows an electron density plot for the Zn$_x$Bi$_2$Se$_3$ skeleton, and Bader analysis reveals that some electrons from Zn are transferred to E-Bi$_2$Se$_3$ after Zn$^{2+}$ intercalation into the E-Bi$_2$Se$_3$ cathode. Consequently, the intercalated Zn$^{2+}$ bears an effective charge of 1.89 (smaller than the nominal charge of 2), contributing to the ions' high mobility in E-Bi$_2$Se$_3$[60].

To elaborate the temperature-dependent conductance characteristics of the topological electrode, several E-Bi$_2$Se$_3$ and Zn$_x$Bi$_2$Se$_3$ nanosheets, obtained by discharging the E-Bi$_2$Se$_3$ cathode to 0.1 V at 0.3 A g$^{-1}$, are separately configured for direct-current transport measurements by using standard electron beam lithography and thermal evaporation of Cr/Au contacts as displayed in the inset of I-V curves of Zn$_x$Bi$_2$Se$_3$ (Fig. 6e). The corresponding current versus 1000/T plots over the temperature range −195 to 25 °C for the E-Bi$_2$Se$_3$ and Zn$_x$Bi$_2$Se$_3$ devices are presented in Supplementary Fig. 29a and Fig. 6f, respectively. The linear I–V relationship proves that the ohmic-like contacts between the Zn$_x$Bi$_2$Se$_3$ nanosheets and Cr/Au electrodes[61]. Notably, the resistance of the as-prepared E-Bi$_2$Se$_3$ nanosheets and discharging-product Zn$_x$Bi$_2$Se$_3$ nanosheets decrease as the temperature declines from 25 to −195 °C, in favorable agreement with the expected metallic behavior. Here, the effect of heavily doped properties of bulk defects is not ruled out. When the temperature drops, the current of the Zn$_x$Bi$_2$Se$_3$ nanosheet device steadily rises, indicating a continued reduction in resistance; the decrease in resistance of E-Bi$_2$Se$_3$ tends to saturation, and the resistance of Zn$_x$Bi$_2$Se$_3$ is much lower than that of E-Bi$_2$Se$_3$, demonstrating the metallic temperature dependence of both materials and prominently improved conductivity (~1.2–1.5 times) after Zn$^{2+}$ embedding in Zn$_x$Bi$_2$Se$_3$ nanosheets.

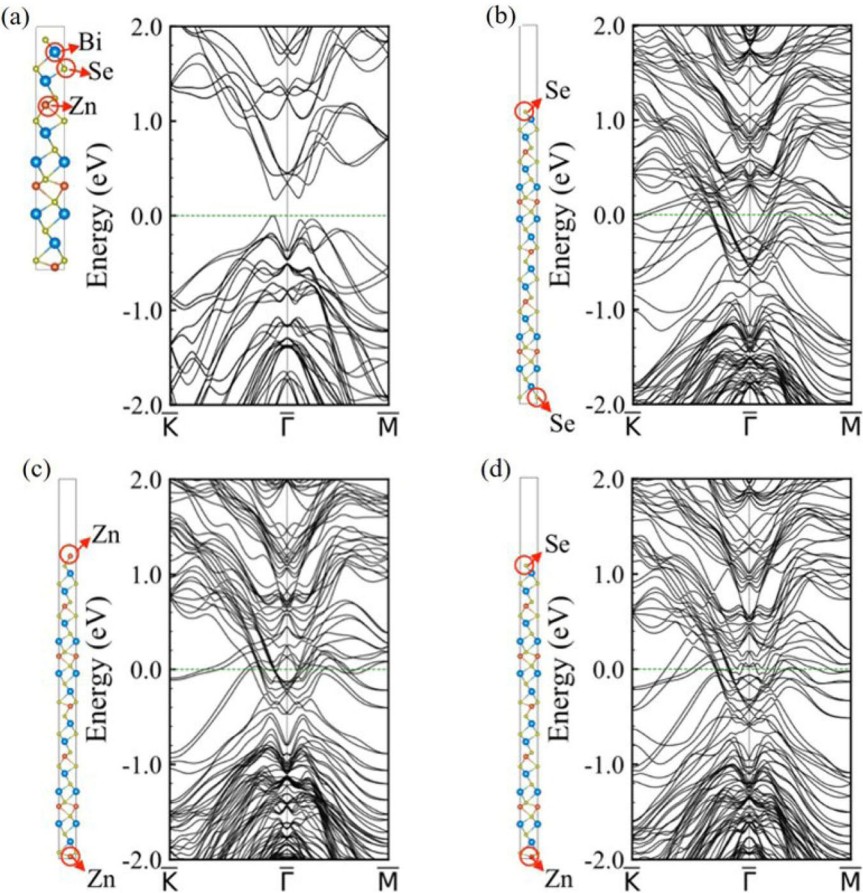

**Fig. 7 Theoretical calculations of spin-orbit coupling (SOC) band structure of Zn²⁺ intercalated in Bi₂Se₃ interlayers. a** Illustration of the supercell for modeling Zn²⁺ intercalated in Bi₂Se₃ interlayers (here denoted as Zn²⁺Bi₂Se₃) and the correspondent SOC band structure. **b–d** SOC band structures of Zn²⁺Bi₂Se₃ 6-QL slab model with Se-Se (**b**), Zn-Zn (**c**), and Se-Zn (**d**) surface termination.

**Spin-orbit coupling band structure calculation of ZnₓBi₂Se₃.**
The information from the electronic structures of these materials is vital to multivalent chemistries[62]. Figure 7a shows the calculated spin-orbit coupling (SOC) band structure of Zn²⁺ intercalated in Bi₂Se₃ interlayers (Zn²⁺Bi₂Se₃), which gives a band gap of 0.17 eV, illustrating the semiconducting nature in the bulk. The 6-QL slab models with different type of elements terminated at the top and bottom surface are employed to investigate the surface electronic structure of the nanosheet (Fig. 7b–d). For all the Se-Se, Zn-Zn, and Se-Zn terminated slabs, the band gap close and make the system metallic, and the band structures do not show any noticeable difference. Compared with the bulk band structure, the conduction bands shift downward and mix with the valence bands, indicating the existence of metallic surface states. In addition, because Bi₂Se₃ has the intrinsic topological surface states (Supplementary Figs. 29b and 30a), even if the topological surface states still exist after Zn²⁺ incorporation, the number of bands will remain unchanged. The metallic surface states play a leading role in the material's conductivity. Due to the overlapping of the conduction and valence bands, it is difficult to distinguish the topological states even if the topological surface states of Zn²⁺Bi₂Se₃ are still retained. The neutral system with intercalated charge-neutral zinc atoms is also calculated for comparison (see Supplementary Fig. 30b–e for details), which implies the existence of topological surface states. It can be concluded that after Zn²⁺ intercalated into the E-Bi₂Se₃ cathode, Zn²⁺ tends to be located around Se atoms on one side, elongating the Bi-Se bonds and resulting in an increased lattice parameter along the *c* axis, as illustrated in Supplementary

Fig. 31. A bandgap exists in the bulk, and the results obtained through the calculation for the 6-QL slab model display surface metallicity; the conductivity of ZnₓBi₂Se₃ benefits from metallic surface states, as proposed in the schematic plot of Supplementary Fig. 32.

The dominant mechanisms responsible for the abnormal low-temperature Zn storage performance of Zn∥E-Bi₂Se₃ can be briefly summarized as follows: the surface metallic feature and topological protection result in enhanced electrical conductivity of E-Bi₂Se₃ at low temperatures. In addition, weak lattice vibration of the E-Bi₂Se₃ bilayer at low temperature results in rapid Zn²⁺ diffusion. The highlights of the previous discussion are that electronic and ionic conductivities are improved at low temperatures, accelerating Zn²⁺ (de)intercalation reactions. Additionally, the decent ionic conductivity of the anti-freeze HC-EGPAM electrolyte also contributes to the enhanced performance. It should be noted that while HC-EGPAM provide a good platform to study the unique features of the topological insulating E-Bi₂Se₃ electrode, the high concentration of LiTFSI (21 m) in the electrolytes will inevitably increase the cost of the cell. Alternatives can be developed in future to address this problem[63,64]. The surface metallic feature and topological protection result in enhanced electrical conductivity open a venue for low temperature cells to achieve enhanced performances. Other materials such as topological semimetals with high electrical conductivity and suitable carrier density near Fermi level are enticing candidate materials for low-temperature electrochemical energy storage meriting from their topologically protected surface states.

## Discussion

All cell systems suffer from performance loss or failure at low temperatures, which is a long-standing problem. Herein, we demonstrate a low-temperature cell based on a few-layer $Bi_2Se_3$ topological insulator cathode, Zn anode, and HC-EGPAM electrolyte. The dependence of cell capacity on temperature is unusual—the lower the temperature is, the better the cell performance is. This is different from all previously reported low-temperature cells. The $Zn||E-Bi_2Se_3$ cell delivers remarkable capacities of 327 mAh g$^{-1}$ (25 °C) and 524 mAh g$^{-1}$ (−20 °C) at 0.3 A g$^{-1}$, and decent cycling performance, with 94.6% of its capacity retained over 2000 cycles at 0 °C and a CE approaching 100%. The mechanism underlying the unusual temperature-dependent cell performance is interpreted by cooperatively utilizing experimental and theoretical strategies. It is revealed that, at a lower temperature, $E-Bi_2Se_3$ realizes higher electrical conductivity, which is ascribed to a coupling advantage in the topological surface states. In addition, more rapid Zn ion diffusion in the $E-Bi_2Se_3$ is observed and is attributed to weaker lattice vibration of the $E-Bi_2Se_3$ bilayer at lower temperature. In particular, the discharging-product $Zn_xBi_2Se_3$ exhibits higher conductivity than $E-Bi_2Se_3$, taking advantage of reinforced trivial metallic surface states because of much-enlarged interlayer spacing and structure distortion after $Zn^{2+}$ intercalation. These aspects, together with the prominent anti-freeze capability of the developed HC-EGPAM electrolyte, result in better cell performance at lower temperatures. Our study indicates that use of a topological insulator as an electrochemical electrode may result in substantially enhanced low-temperature cell performances, even better than that at room temperature. The developed cells can be an excellent choice for powering systems that operate in cold areas for long periods. Undoubtedly, this research will inspire research on low-temperature cells from the perspective of using a topological insulator as an electrode.

## Methods

**Preparation of anti-freeze hydrogel electrolyte with high concentration of salts and ethylene glycol (HC-EGPAM).** $Zn(TFSI)_2$ (99% purity) and LiTFSI (99% purity) are purchased from Macklin reagent Co., LTD. The salts are used as received without further purification. A high concentration salt electrolyte is prepared by adding 1 m $Zn(TFSI)_2$ and 21 m LiTFSI [where m is molality (mol kg$^{-1}$) and TFSI denotes bis(trifluoromethanesulfonyl)imide] to 2 mL of deionized water under vigorous stirring at room temperature for 30 min. After the salts are fully dissolved in the water, 2 mL of ethylene glycol (EG) antifreeze is mixed in. Acrylamide, used as the monomer, 0.05 g of ammonium persulfate, used as the initiator, and 0.005 × g of bisacrylamide, used as the crosslinker, are added together and stirred until a transparent dispersion is obtained. EG (99.5% purity), acrylamide (99% purity), ammonium persulfate (99% purity), and bisacrylamide (99% purity) are purchased from Macklin reagent Co., LTD and used as received. Subsequently, free-radical polymerization proceeds in an oil bath at 70 °C for 3 h. For comparison, the polyacrylamide-based hydrogel electrolyte (HC-PAM) is prepared by replacing the EG additive with an equal amount of deionized water in the aforementioned process. 70% EG/water has the lowest freezing point of −55 °C. In our highly concentrated salts system, EG content can be up to 50%.

**Synthesis of few-layer $Bi_2Se_3$ nanosheets (E-$Bi_2Se_3$).** E-$Bi_2Se_3$ nanosheets are readily fabricated using a facile two-step high-energy mechanical milling (HEMM)/ lithium intercalation approach, and $Bi_2Se_3$ powder (P-$Bi_2Se_3$) obtained through HEMM can be easily exfoliated into a stable suspension by using the hydration force (Supplementary Fig. 1). High-purity bismuth shot (Bi, 99.999%; Alfa Aesar) and selenium shot (Se, 99.999%; Alfa Aesar), weighed according to the stoichiometric ratio of $Bi_2Se_3$ are loaded into a stainless-steel ball-milling jar in a glove box at room temperature under argon atmosphere (<1 ppm of $H_2O$ and $O_2$). The jar is subjected to ball milling for 20 h at 1200 rpm. Afterward, 150 mg of ball-milled powder is added to a 70-mL EG solution of lithium hydroxide (8 g L$^{-1}$) under continuous stirring for 30 min, after which the mixture is placed into a 100-mL Teflon-lined autoclave. The autoclave is heated at 200 °C for 28 h to obtain the fully exfoliated $Bi_2Se_3$ by plating/stripping Li$^+$ dispersed in the solution. The dispersion is centrifuged and then washed thoroughly using acetone (the amount of volume used is 250 mL) and deionized water (300 mL) six times. Lithium hydroxide (99.7% purity) and acetone (>99.5% purity) are purchased from Macklin reagent Co., LTD and used as received. A small amount of $SeO_2$ impurity introduced during the HEMM process can be removed after washing with the tetramethylammonium hydroxide (TMAH, 0.023 M)/NaOH (0.006 M)/NaCl (0.016 M) aqueous solution

and deionized water (Supplementary Fig. 7)[30]. TMAH (25 wt.% in $H_2O$), NaOH (68 g/L in $H_2O$) and NaCl (>99.5% purity) are purchased from Macklin reagent Co., LTD and used as received. After vacuum filtering through porous poly-vinylidene fluoride (PVDF) membranes of 0.45-mm nominal pore size, the resulting samples are dried in vacuum at 40 °C overnight.

**Fabrication and electrochemical characterization of the quasi-solid $Zn||E-Bi_2Se_3$ cell.** To prepare the cathode, E-$Bi_2Se_3$, acetylene black (>99.99% purity, Macklin reagent), and PVDF adhesive (Solvay 5130 with a purity > 99.5%, Solvay Specialty Polymers) are mixed uniformly in a 7:2:1 weight ratio with N-methyl-2-pyrrolidone (NMP, AR, Aladdin Reagent); then, the slurry is cast on a carbon cloth with a thickness of 1 mm (WOS 1009, Cetech Co., LTD), which is then vacuum dried. The areal loading of E-$Bi_2Se_3$ is approximately 1.6 mg cm$^{-2}$. Electrodeposited Zn on carbon cloth (in flexible cells for demonstration) or Zn foil with purity of 99.99% and thickness of 0.15 mm purchased from Shanghai Ailiai Metallic Material Co., LTD (in CR2032 coin-type cells to evaluate the electrochemical properties) is used as an anode. The coin cell is crimped with a pressure of 5 MPa. The quasi-solid $Zn||E-Bi_2Se_3$ cell is assembled by sandwiching HC-EGPAM between the Zn anode and E-$Bi_2Se_3$ cathode. CVs and EIS are obtained using a CHI 760E workstation. EIS is conducted at a quasi-stationary potential with two-electrode system in the frequency range from 1 MHz to 0.1 Hz at open-circuit voltage, 5 mV amplitude. The recording number of data points is 12 per decade. GCD profiles, rate capability, and cycling performance are determined using the LAND testing system. In order to avoid the influence of cell activation, the order of measurement of temperature-dependent rate ability is inititated from a small specific current of 0.3 A g$^{-1}$ to a large specific current of 5 A g$^{-1}$. The test begins after the completion of activation at 0.3 A g$^{-1}$ at 25 °C (around three cycles). The ionic conductivity (σ) of the hydrogels can be calculated from alternating-current impedance spectra obtained using two stainless-steel planar electrodes sandwiching the hydrogels. Then, σ is calculated as a function of the ohmic resistance (R; namely the X-axis intercept of the profile), thickness (l), and test area of the hydrogel (A) through the following equation:

$$\sigma = \frac{l}{RA} \qquad (1)$$

**Material characterization.** XRD data are collected using a Bruker D2 Phaser diffractometer and Cu-Kα radiation (λ = 0.154 nm). XRD refinement is performed using the GSAS-EXPGUI program. The morphology of the electrodes is examined using TEM (JEOL-2001F) and field-emission SEM (FEI/Philips XL30). Inductively coupled plasma-atomic emission spectroscopy (ICP-AES) is performed using the PerkinElmer Optima 8300. Raman spectroscopy is conducted using a multichannel modular triple Raman system (WITec alpha300 access). Chemical state and composition are analyzed using XPS (ESCALAB 250 photoelectron spectroscopy). For ex situ measurements of the cathode, batteries are firstly charged/discharged up to the specific potential using a LAND workstation. At the end of charged/discharge, these batteries are opened in the air to collect cathodes. Afterward, these cathodes are washed with deionized water three times. Finally, cathodes are dried in an oven at 60 °C for 20 min for further ex situ measurements. For electrical transport experiments on the as-fabricated E-$Bi_2Se_3$ and discharging-product $Zn_xBi_2Se_3$, shadow masking is used to define the drain and source regions. Cr/Au film electrodes of thickness 5/80 nm are deposited using electron beam evaporation. The channel lengths are 10 μm for all devices. The devices are characterized using a standard electrical probe station and Agilent 4155C semiconductor analyzer in an ambient atmosphere. The low temperature test is done in ESPEC high and low temperature test chamber (GPS-3).

**Calculation.** The DFT calculations are performed using the Vienna ab initio simulation package (VASP)[61] with the Perdew-Burke-Ernzerhof-type generalized gradient approximation (GGA-PBE)[65]. The electron-ion interaction is described by projected augmented wave (PAW) method[66]. The cutoff energy for the plane-wave basis sets is set to be 350 eV. The spin-orbit coupling (SOC) is taken into account due to the strong relativistic effect in Bi element. A slab model contained 6-QLs with a vacuum region of more than 15 Å is used to model the E-$Bi_2Se_3$ nanosheets. The Brillouin zone is sampled by a 12 × 12 × 2 and 12 × 12 × 1 Γ-centered k-mesh for bulk and slab model, respectively. All structures are completely relaxed until the residual force on each atom is <0.01 eV/Å. The Grimme's DFT-D3 method is used to describe the van der Waals interactions[67]. In order to simulate the $Zn^{2+}$ diffusion process in the E-$Bi_2Se_3$ interlayers, we perform the ab initio MD (AIMD) simulations using the Nosé algorithm[68], with a time step of 2 fs and a simulation period of 10 ps. A 4 × 4 × 1 supercell containing 161 atoms and a k-point sampling at Γ-point is used in the AIMD.

**Reporting summary.** Further information on research design is available in the Nature Research Reporting Summary linked to this article.

## Data availability

The data that support the findings of this study are available within the text including the Methods, and Supplemental information. Raw datasets related to the current work are available from the corresponding author on reasonable request.

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

## Acknowledgements

This research was supported by the National Key R&D Program of China (no. 2019YFA0705104 (C.Z.)). The work was also partially sponsored by GRFs under Project CityU 11305218 (C.Z.), CityU 11212920 (C.Z.), and the Guangdong Innovative and Entrepreneurial Research Team Program (no. 2016ZT06G587 (W.L.)). The authors would like to thank Mr T. F. Hung for HRTEM analysis.

## Author contributions

Yw.Z.: conceptualization, methodology, and writing. Y.L.: methodology. H.L.: methodology. Yb.Z.: methodology. Y.M.: methodology. N.L.: methodology. D.W.: methodology. F.J.: methodology. F.M.: methodology. C.L.: methodology. Y.G.: methodology. X.L.: methodology. Z.H.: methodology. Q.L.: methodology. J.C.H.: methodology. J.F.: methodology. M.S.: methodology. F.C.: methodology, review, and editing. W.Z.: methodology, review & editing. W.L.: methodology, review, and editing. C.Z.: conceptualization, funding acquisition, writing – review and editing.

## Competing interests

The authors declare no competing interests.
