## [Peer Review File · Nature Communications]

REVIEWER COMMENTS

Reviewer #1 (Remarks to the Author):

The pressing demand for high energy availability at lower costs and better sustainability has been driving the quest for developing the technology beyond Li-ion batteries. Zn-ion battery represents one of the most promising candidates for energy storage. This work reports the use exfoliated Bi₂Se₃ nanosheets for storage of Zn²⁺, particularly at lower T (-20°C). The materials show improved performance when lowering the T, which the authors attribute to the increased effect of surface conducting-nature of the Bi₂Se₃ nanosheets when their thickness is only a few layers. Although new insights are presented with the multiscale characterizations, the practicality of this topological insulator that was rarely used in intercalation-type battery due to its intrinsic nature remain questionable. Also, the mechanism discussion regarding Zn²⁺ insertion and capacity increase at lower temperature is very weak. Based on above concerns, I doubt that the present work is of enough significance to justify publication in Nature Communications and suggest submitting to a more specialized journal.

1. Frankly speaking, the practicality of the few-layer E-Bi₂Se₃ remains questionable. It may lose the advantage when increasing electrode loading or particle size and limiting electrolyte usage. Likewise, such a two-dimensional material will very likely serve as a capacitor or a hybrid of capacitor & battery rather than a pure battery. If a combined mechanism, what proportion each of them accounts for. The story may be changed when capacitor behavior is involved.
2. The mechanism discussion on increasing capacity at lower temperatures is very weak and lacks conclusive experimental evidence.
3. The claim that "this abnormal low-temperature response of battery performance is firstly observed.", which might not be true based on the literature search of the reviewer. For example: Counter-intuitive Structural Instability Aroused by Transition Metal Migration in Polyanionic Sodium Ion Host. Adv. Energy Mater. 2020, 2003256.
4. For the STEM analysis in Figure 1, the identification of Se is reluctant, not convincing. Image is not with high quality. It is hard to believe that figure f is an enlarged view of the image in figure e, which apparently exhibits a poor resolution. In addition, why do the two Se atoms sitting on Bi sides show asymmetric position compared to Bi in the center? Could this be caused by sample drift during imaging capturing?
5. Why the capacity in figure 3d sharply increases in the first tens cycles. Does this capacity increase share the same mechanism with that observed at lower temperature? The authors should explain it in detail.
6. Figure 4j and k also raise serious concerns from the reviewer. One comment is similar to that of Figure 1 f and g, i.e. concerning the quality of Figure 4i in obtaining what shown in Figure 4k. The other comment is about the relative positions of Bi and Se shown in Figure 4k, particularly the 4th (from the left) Se atom, which should be theoretically sitting in the middle of the 2nd Bi and the 3rd Bi (from the left) according to the atomic model, while the line profile shows that Se is apparently not in the middle...The reviewer highly doubts that there is significant sample drift during imaging capturing, and such drift can further cause the artifacts in the contrast analysis, and the authors should clarify this.
7. The EDS mapping in Figure 4h is not convincing for Zn insertion into the lattice of Bi₂Se₃. Apparently, Zn signal has a different distribution compared to that of Se, and Bi signal can be barely seen. If Zn²⁺ indeed insert into the lattice, then the three should have a similar signal distribution.
8. Some descriptions are very misleading. For example, "Even at -40°C, the capacity retentions remain to 106 % and 113 % compared to the capacities at 25 °C." It should not be described as capacity retention because the final materials are changing with the decreased temperature.
9. All coulomb efficiency figures should be modified to a narrow percentage range.

Reviewer #2 (Remarks to the Author):

This manuscript by Zhao et al. reports an aqueous Zn/Bi₂Se₃ system in an antifreeze and concentrated electrolyte. Very surprisingly, in a temperature range of 25 - -20 °C, as the temperature decreases, the performance of the battery becomes even better. This has never been observed in any battery system: all battery will have capacity degradation at lower temperature. Bi₂Se₃ as a topological insulator contribute to the abnormal behavior. The Zn/ Bi₂Se₃ delivers high capacity and energy density at -20 °C (524 mAh g⁻¹ and 441 Wh kg⁻¹). Also, the mechanism explanations are solid with many characterizations to support. The improved performance is due to the enhanced contribution of surface topological electronic states and weaker lattice vibration of the few-layer topological insulator bismuth selenide at lower temperature. In general, this is a manuscript with high novelty. Hence, I recommend the manuscript to be published with minor revisions addressed.

1. For the topological insulators, the surface states after exfoliating are enhanced and the electrical conductivity is also increased. Does this indicate that whether the material itself can be exfoliated is a key factor for the electrochemical performance in the selection of topological insulators?
2. On page 3, "...is formed by a periodic layers", and on page 4 "... the kinetics of the battery reactions at lower temperature.", and on page 8 "Thickness of numerous flakes is calculated...". Pay attention to the use of articles in the paper. Incorrect use of articles can also lead to confusion relating to singular vs plural senses.
3. The role of ethylene glycol in antifreeze electrolytes is critical. It is recommended to explain the role of ethylene glycol and the factors to consider in the selection of ethylene glycol content.
4. In Fig 5g-j, when Zn²⁺ intercalated in Bi₂Se₃ interlayers (Zn²⁺+Bi₂Se₃), is it still a topological insulator.
5. I think this paper may open a new venue for low temperature batteries, can the authors discuss more on what electrodes may have this unique feature other than topological insulators?

Reviewer #3 (Remarks to the Author):

These authors report on the aqueous Zn//Bi₂Se₃ batteries in which the electrolyte is a high-salt/acrylamide gel medium and the cathode is an exfoliated form of Bi₂Se₃ previously reported to show improving conductivity as temperature is lowered. The paper is very detailed in terms of characterization, but the main findings are relatively straightforward: at lower temperature the batteries exhibit higher capacity that at ambient conditions. Authors attribute the effect mainly to materials physics & charge transport in the cathode, which seems reasonable considering that the control cells that use Bi₂Se₃ powder show mostly capacitive behaviors and minimal faradaic currents for the same electrolyte and temperature.

Overall the study is of some interest, but there are at least three important concerns that would make it of limited technical impact. Although these concerns center on the battery performance and not the characterization, they are actually of greater importance because the origins of the improved charge transport in few layer Bi₂Se₃ are already known, so the materials themselves and the detailed analysis in the paper are actually of low novelty. The revised submission needs to clearly address the concerns below to allow this reviewer to understand what is the actual value of the contribution.

Major Concerns

1. The actual capacity of the cathodes used in the battery studies are impractically low (my estimate 0.4mAh/cm²), which would make such Zn batteries not competitive with state-of-the are commercial Zn batteries and impractical relative to Li-ion batteries , regardless of their good low temperature

behaviors. Considering that the EIS results and previous literature indicate that E-Bi₂Se₃ becomes more conductive at low temperature, it would seem that cathode capacities at least one order of magnitude higher (e.g. 2-4mAh/cm²) would be viable. Yet authors do not report results for such systems. Please explain how the cathode capacity was selected? Please also provide results at higher cathode capacity (at least 2mAh/cm²).

2. The high concentration of LiTFSI (21 m) in the electrolytes is also impractically high. Again, such batteries would not be of practical interest or competitive with commercial alkali Zn cells. Considering that the authors also use an antifreeze in the electrolyte, is the high salt concentration really necessary. It will be helpful to the readers to understand how the antifreeze to salt ration influence the low temperature ion transport behaviors in the HC-EGPAM electrolytes. In other word, if as the authors claim, the higher low temperature capacity comes form enhance transport in the cathode, why is so much salt needed in electrolyte?

3. In the abstract and at several places in the body of the manuscript, authors mention that their Zn//E-Bi₂Se₃ batteries can operate to temperatures as low as -50 C. It would be helpful to see the ionic conductivity data for the HC-EGPAM electrolytes (Fig. S9) at lower temperature than those reported now.

In this Response Letter to Referees, we have addressed and clarified all the helpful and valuable comments from referees. With a substantial amount of new results added into the revised manuscript according to the referees' comments, we are very grateful to the reviewers for their dedication to this work.

Below, we provide our point-by-point responses to the referees' comments, where the original comments are shown in black and our responses are shown in blue.

Referee #1:

The pressing demand for high energy availability at lower costs and better sustainability has been driving the quest for developing the technology beyond Li-ion batteries. Zn-ion battery represents one of the most promising candidates for energy storage. This work reports the use exfoliated Bi_2Se_3 nanosheets for storage of Zn^{2+} , particularly at lower T (-20 °C). The materials show improved performance when lowering the T, which the authors attribute to the increased effect of surface conducting-nature of the Bi_2Se_3 nanosheets when their thickness is only a few layers. Although new insights are presented with the multiscale characterizations, the practicality of this topological insulator that was rarely used in intercalation-type battery due to its intrinsic nature remain questionable. Also, the mechanism discussion regarding Zn^{2+} insertion and capacity increase at lower temperature is very weak. Based on above concerns, I doubt that the present work is of enough significance to justify publication in Nature Communications and suggest submitting to a more specialized journal.

Response: Thank you very much for your comments. We admit that, in terms of technology maturity, zinc ion batteries are still much behind lithium-ion batteries. There will be many uncertain factors to realize practical applications of the developed Bi_2Se_3 cathode.

However, the main message of this paper is to report the possibility that a battery may have better performance at lower temperature through adopting electrode materials with novel properties. Topological Bi_2Se_3 is only one choice to achieve this performance and many other may exist for us to explore. We believe this is an important progress of low temperature batteries as most work on low temperature batteries focused on electrolyte optimization previously. In this work, we

present that with the support of a low temperature applicable electrolyte, adopting an electrode with unique properties may result in even better results.

These batteries may be used in long-term cold area where the battery must serve at low temperature through the batteries' lifespan. When the practical applications are realized, probably it is not Bi_2Se_3 or a topological insulator, but the idea to use electrode with unique features and suitable electrolytes for low temperature batteries may come from our paper.

Please be also noticed that although zinc ion batteries are not as mature as lithium-ion batteries, they still have great chance to share a certain portion of battery market with its superior safety performance, considering recent many fire accidents induced by lithium-ion batteries.

Please find our detailed response to your specific comments below.

1. Frankly speaking, the practicality of the few-layer E- Bi_2Se_3 remains questionable. It may lose the advantage when increasing electrode loading or particle size and limiting electrolyte usage. Likewise, such a two-dimensional material will very likely serve as a capacitor or a hybrid of capacitor & battery rather than a pure battery. If a combined mechanism, what proportion each of them accounts for. The story may be changed when capacitor behavior is involved.

Response:

We appreciate your insightful comments. We also agree that the goal of scientific research should be practicality and admit there is a long way to go for practical application of the developed batteries. When a battery with better performance at lower temperature is commercialized, it may be not Zn// Bi_2Se_3 , but the idea may originate from our paper to adopt electrode materials with unique features.

Following your suggestion, to evaluate utility of Zn//E- Bi_2Se_3 batteries, E- Bi_2Se_3 cathodes with a mass loading of 1-4 mg cm^{-2} were prepared and tested from 0.1 A/g to 5 A/g (Fig. S14). Fig. S14a plots the areal capacity vs. the areal mass loading of the electrodes at -20 °C. The areal capacity increases linearly with the increasing areal mass loading at current densities $<3 \text{ A g}^{-1}$. When the current density is larger than 3 A g^{-1} , the dependence slightly deviates from the linear relationship. Fig. S14b shows the areal capacity as a function of the areal current density in which the E- Bi_2Se_3

cathode with a mass loading of 4 mg cm⁻² exhibits areal capacities of 2.1, 1.84, 1.52, 1.17, 0.72, and 0.46 mAh cm⁻² at the areal current density of 0.4 (0.1 A/g), 1.2 (0.3 A/g), 2 (0.5 A/g), 4 (1 A/g), 12 (3 A/g), and 20 (5 A/g) mA cm⁻², respectively. Temperature-dependent areal capacity of E-Bi₂Se₃ cathodes at 0.3 A/g are displayed in Fig. S14c and Fig. S14d, respectively. The E-Bi₂Se₃ cathode with a mass loading of 4 mg cm⁻² delivers areal capacities of 1.84, 1.45, 1.14, 1.01 and 0.89 mAh cm⁻² at the temperature of -20, -10, 0, 10, and 20 °C, respectively. Please see Page 14 in the revised manuscript and Page 16 (Fig. S14) in the revised supplementary information.

Understanding the operative mechanisms of energy storage systems is extremely important. We follow your suggestions and calculate the diffusion and capacitive contribution of our battery at a series of scan rates (Fig. S17). According to Equation (1)¹, the b values of peak 1 and 2 are 0.81 and 0.72, respectively, demonstrating that presence of a synergistic charge storage process. Both diffusion-controlled and capacitive behaviors in Zn//Bi₂Se₃ batteries are responsible for the fast kinetics during the discharge/charge process. To further specify the capacitive contribution at a certain scan rate, the following Equation (2) is employed.² The result shows that proportion of ≈ 73-8% of the whole capacity originate from the diffusion contribution at scan rates of 0.1-5 mV s⁻¹.

$$i = av^b \quad (1)$$

$$i = k_1v + k_2v^{1/2} \quad (2)$$

These ratios are normal in the common zinc ion batteries.

At all current densities with different ratios of diffusion and capacitive contribution, we observe better performance at lower temperatures. The battery at low temperature follows the same mechanism with that at room temperature.

For better clarity, we have added some explanations and discussions in the revised supplementary information (Fig. S17) in this regard. Please see Page 21 in the revised supplementary information.

1. Nat Mater 16, 454-460 (2017).
2. Electrochemical Methods: Fundamentals and Applications 2nd edn (Wiley, 2001).
3. Science 343, 1210-1211 (2014).

2. The mechanism discussion on increasing capacity at lower temperatures is very weak and lacks conclusive experimental evidence.

Response:

Thank you for your valuable comment. Following your suggestion, we further reinforce the investigation and explanation. The battery at low temperatures follows the same Zn storage mechanism with that at room temperature. The enhanced capacity at lower temperatures benefits from both high electronic conductivity and high ionic conductivity.

1. The enhancement of electronic conductivity of E-Bi₂Se₃ at lower temperatures due to topological surface states can be illustrated by electrochemical impedance spectra (EIS) of Zn//E-Bi₂Se₃ batteries and electrical transport experiment of Bi₂Se₃ and Zn_xBi₂Se₃. First, the slightly increased R_s of Zn//E-Bi₂Se₃ from 6.2 Ω at 20 °C to 8.55 Ω at -20 °C, which includes the resistance of the HC-EGPAM, separator, and E-Bi₂Se₃ electrode, reflects the much better electronic conductivity of the batteries at low temperatures. Second, the corresponding current versus 1000/T plots over the temperature range -195 to 25 °C for the E-Bi₂Se₃ and Zn_xBi₂Se₃ demonstrates that the conductance increases as the temperature decreases.

2. The more rapid Zn ion diffusion of the E-Bi₂Se₃ at lower temperatures can be attributed to weaker lattice vibration of the E-Bi₂Se₃ bilayer, which is illustrated by electrochemical impedance spectra (EIS) of Zn//E-Bi₂Se₃ batteries and diffusivity coefficient (D) of Zn²⁺ of Bi₂Se₃ and Zn_xBi₂Se₃ calculated using the GITT. The average Zn²⁺ diffusion coefficient of the Zn//E-Bi₂Se₃ battery is calculated to be 10⁻¹⁰-10⁻¹¹ cm² s⁻¹, higher than that of the Zn//P-Bi₂Se₃ battery. The faradic impedance (namely the combination of charge-transfer resistance (R_{ct}) and Warburg impedance (Z_W)) reflects the kinetics of the battery reactions. Warburg impedance is assigned to the Zn²⁺ diffusion in the battery.⁴ The slope is proportional to the Z_W .⁵ Zn//E-Bi₂Se₃ shows much faster Zn²⁺ diffusion coefficient. Herein, the considerably lower R_{ct} at -20 °C (288 Ω) of Zn//E-Bi₂Se₃ than the R_{ct} of Zn//P-Bi₂Se₃ (500 Ω) favorably escorts fast Zn²⁺ diffusion kinetics. These results reveal that the stable ionic transport and superior electric conductivity of the Zn//E-Bi₂Se₃ battery in cold environments.

3. Spin-orbit coupling (SOC) band structure of Zn²⁺ intercalated in Bi₂Se₃ interlayers (Zn²⁺Bi₂Se₃), which gives a band gap of 0.17 eV, illustrating the semiconducting nature in the bulk. For all the

Se-Se, Zn-Zn, and Se-Zn terminated slabs of the 6-QL slab $\text{Zn}^{2+}\text{Bi}_2\text{Se}_3$ models, the band gap close, making the system metallic. Compared with the bulk band structure, the conduction bands shift downward and mix with the valence bands, indicating the existence of metallic surface states. It can be concluded that after Zn^{2+} intercalated into the E- Bi_2Se_3 cathode, Zn^{2+} tends to be located around Se atoms on one side, elongating the Bi-Se bonds and resulting in an increased lattice parameter along the c axis. A bandgap exists in the bulk, and the results obtained through the calculation for the 6-QL slab model display surface metallicity. The conductivity of $\text{Zn}_x\text{Bi}_2\text{Se}_3$ benefits from metallic surface states.

4. Then we compare our topological insulator E- Bi_2Se_3 electrode with Prussian blue analogue and MnO_2 electrodes in the same antifreeze hydrogel electrolyte⁶⁻⁷ (Fig. R1). Considering that the performance of other electrode materials remarkably decreases with the decrease of temperature, we confirm that the exfoliation-enhanced topological surface state of the E- Bi_2Se_3 is the key to achieve the improved performance at low temperature. Of course, to obtain superior low-temperature performance, it is essential to ensure that the electrolyte is in a good working condition at low temperature.

The dominant mechanisms responsible for the abnormal low-temperature Zn storage performance of Zn//E- Bi_2Se_3 can be briefly summarized as follows: the surface metallic feature and topological protection leads to enhanced electrical conductivity of E- Bi_2Se_3 at low temperatures. In addition, weak lattice vibration of the E- Bi_2Se_3 bilayer at low temperature results in a rapid Zn^{2+} diffusion. Moreover, the decent ionic conductivity of the anti-freeze HC-EGPAM electrolyte also contributes to maintain a good battery performance.

Fig. R1 Temperature-capacity retention dependence of E- Bi_2Se_3 and other cathode materials-based zinc ion batteries in the antifreeze electrolytes⁶⁻⁷.

4. *Electrochimica Acta* 51, 1376-1388 (2006).
 5. *Electrochimica Acta* 48, 79-84 (2002).
 6. *Adv. Mater.* 1901521 (2019).
 7. *Energy Environ. Sci.* 12, 706-715 (2019).
3. The claim that “this abnormal low-temperature response of battery performance is firstly observed.”, which might not be true based on the literature search of the reviewer. For example: Counter-intuitive Structural Instability Aroused by Transition Metal Migration in Polyanionic Sodium Ion Host. *Adv. Energy Mater.* 2020, 2003256.

Response:

We regret for this oversight and thank the referee for bringing it to our attention. We have revised our statement to be more precise and cited the paper. However, please be noted that the phenomenon reported in the *Adv. Energy Mater.* 2020, 2003256 is because the instability of NVCP, that is, they obtained a poorer performance at higher temperature. Please see the following sentences directly copied from the paper:

“We have realized 1.5-electron $V^{3+}/V^{4+}/V^{5+}$ multielectron reaction of NVCP and observed irreversible charge/discharge curves as well as quick capacity lost when cycling that at 30° C...”

“.... a strategy for recovering the structure of NVCP at 30° C will be proposed..”

“This indicates that the electrochemical reactions at higher temperatures tend to be irreversible compared with those at lower temperature...”

4. For the STEM analysis in Figure 1, the identification of Se is reluctant, not convincing. Image is not with high quality. It is hard to believe that figure f is an enlarged view of the image in figure e, which apparently exhibits a poor resolution. In addition, why do the two Se atoms sitting on Bi sides show asymmetric position compared to Bi in the center? Could this be caused by sample drift during imaging capturing?

Response:

Thank you for your valuable comment. We accepted review's comment on the quality of our HAADF image. Our E-Bi₂Se₃ sample is very beam sensitive, it is better to image at low voltage and at low dose mode to minimize radiation damage from electron beam and therefore to improve the image quality. The current image is recorded in our aberration corrected TEM operated at 300 keV which may degrade crystallinity of our sample. Here, we intend to improve the visibility of Se-Bi-Se via a set of signal analysis.

In order to confirm the data consistency in Fig. 1, the low magnified and enlarged view of HAADF images are presented in Fig. R3a and R3b, respectively, from which we could identify the atomic configuration of E-Bi₂Se₃. In ideal conditions, the distance between Se-Bi-Se columns is 4.23 Å, and Bi atom sites symmetrically between two Se atoms (the Schematic in Fig. R3b). We averaged several plots across Se-Bi-Se cluster along pink dashed lines. Assume the atom is a Gaussian shape. The averaged plot (average of 8 line profiles) is fitted with Gaussians. The spacing between two Se-Se along the dash line from fitted Gaussians is 4.14Å with the experimental error of 0.3Å. Therefore, the confirmation of existence of Se-Bi-Se is reasonably in good confidence.

However, as fitting the peak intensity of line profile in Fig. R3c, two Se atoms do not distribute symmetrically around Bi site (the distance of Se-Bi at the left side is 0.248 nm, and the distance of Bi-Se at the right side is 0.171 nm, which is an obvious difference as considering the experimental error of 0.03 nm). To interpret the phenomena, the reviewer here proposes an assumption that sample drift during imaging capturing may play a critical role. However, the atomic configuration in the HAADF image could be identified clearly, which is apparently with no image drift at all in Fig. R3. Normally, ball-milling and hydrothermal stripping could affect the crystallinity greatly during the sample preparation. Especially, it is well-known that the electron beam knock-on may displace the atoms in crystal.⁸ As capturing the atomic-scaled HAADF image of Fig. R3 along [010] zone axis, the atomic configuration at different areas may have certain distortion, and this is why we could observe an unsymmetrical Bi atom sitting between two Se atoms (Fig. R3).

Once again, we thank the referee for this comment. We have provided original and high-resolution HAADF-STEM images in Fig. 1 and Fig. S5 and added the explanation in the revised manuscript (Page 7 and 8) and supplementary information (Page 6 and 7).

Fig. R3 (a) High-resolution HAADF-STEM image of E-Bi₂Se₃ (b) Schematic of the atomic configuration of Bi and Se atom along [010] projection, the green sphere presents the Se atom and purple sphere indicates the Bi atom. The distance of two Se atoms as indicated is 4.23 Å while the spacing of two Se is 4.17 Å. Such atomic configuration fits well with the enlarged HAADF image. (c) Average line intensity profile along the HAADF image (the pink dash line at the bottom of HAADF image), which is cut up from the bottom area of HAADF image in (b).

8. Ultramicroscopy 127, 100-108 (2013).

5. Why the capacity in figure 3d sharply increases in the first tens cycles. Does this capacity increase share the same mechanism with that observed at lower temperature? The authors should explain it in detail.

Response:

We thank referee for this constructive comment. The capacity in Fig. 3d sharply increases in the first tens cycles is due to the activation of the electrode. Hydrogel electrolyte cannot completely infiltrate the E-Bi₂Se₃ in the first cycle, which results in the gradual capacity increase over quite a few cycles. In order to avoid the influence of battery activation, the order of measurement of temperature dependent rate capability is initiated from a small rate of 0.3 A/g to a large rate of 5 A/g. The test begins after the completion of activation at 0.3 A/g at 25 °C (around 3 cycles).

We have revised our manuscript for better clarify, please see Page 13 and 26 in the revised manuscript. Thanks again for your considerate advice.

6. Figure 4j and k also raise serious concerns from the reviewer. One comment is similar to that of Figure 1 f and g, i.e. concerning the quality of Figure 4i in obtaining what shown in Figure 4k. The other comment is about the relative positions of Bi and Se shown in Figure 4k, particularly the 4th (from the left) Se atom, which should be theoretically sitting in the middle of the 2nd Bi and the 3rd Bi (from the left) according to the atomic model, while the line profile shows that Se is apparently not in the middle...The reviewer highly doubts that there is significant sample drift during imaging capturing, and such drift can further cause the artifacts in the contrast analysis, and the authors should clarify this.

Response:

Thanks for your insightful comments. We are sorry for our oversight that the Zn and Se at around 2.0 nm were not marked in the original Fig. 4k. We now mark the positions as shown in the Fig. R4a and b. The structure perfectly shows the Bi atoms are in the middle of two Se atoms and Zn is intercalated into the lattice.

To further confirm the conclusion, we performed another HAADF test for the $Zn_xBi_2Se_3$ sample and the results are shown in Fig. R4c-f. The high image quality indicates there is almost no sample drift presents here. In order to confirm the symmetry of two Se atoms next to the Bi atom site, the average line scan across the Se-Bi-Se columns is draw in Fig. R4f. Here the distance of Se to Bi atoms are 0.209 nm and 0.217 nm, respectively, and the measurement error is about 0.03 nm. Therefore, the Bi atom is apparently in the middle position between two Se atoms. However, with the same issues we have discussed in Fig. R3, the distortion of lattice at other areas may induce the deflection of Se-Bi-Se columns along [010] zone axis. The discussions have been added in the supplementary information (Fig. S22 on Page 27).

Once again, we thank the referee for this comment. The original Fig. 4j and k are put in the revised supplementary information (Fig. S22c and d). The Figure 4k and 4j has been updated. Please see new Figure 4i and 4j in the revised manuscript.

Fig. R4 (a) A magnified view with the superposition of the E-Bi₂Se₃ crystal structure. The yellow line shows (1 0 10) crystallographic plane with the $[\bar{1}001]$ crystallographic direction. (b) An averaged intensity profile from four traces of the (1 0 10) crystallographic plane in (a). The power law between (I_{Bi}/I_{Se}) and (Z_{Bi}/Z_{Se}) is analyzed with an averaging of three Se-Bi-Se peaks. (c) HAADF-STEM image of Zn_xBi₂Se₃ and (d) its corresponding fast fourier transform (FFT) pattern. (e) High-resolution HAADF-STEM image of Zn_xBi₂Se₃ in (c), which confirms that the HAADF image is captured along [010] zone axis. (f) Magnified view of the red square area in (a). (d) Average line scan of the intensity profile (the peach dash line at the bottom of the HAADF image).

7. The EDS mapping in Figure 4h is not convincing for Zn insertion into the lattice of Bi₂Se₃. Apparently, Zn signal has a different distribution compared to that of Se, and Bi signal can be barely seen. If Zn²⁺ indeed insert into the lattice, then the three should have a similar signal distribution.

Response:

Thanks for the constructive comments and sorry for the misunderstanding induced. In the revised manuscript, we provide the HAADF-STEM image and corresponding EELS elemental maps of Bi, Se and Zn in Fig. 4k.

In addition, in order to confirm the insertion of Zn^{2+} into the Bi_2Se_3 lattice, quantification for the HAADF image is further discussed in Fig. S22.

The HAADF-STEM images of E- Bi_2Se_3 (Fig. R3d) and $\text{Zn}_x\text{Bi}_2\text{Se}_3$ (Fig. R4c) are taken along the [010] zone axis. As we all known, the HAADF intensity I of one atom is proportional to its atom number Z of n power law $I=Z^n$ in the HAADF model.^{9,10} In order to identify the power law n in this experimental and quantify the HAADF image of Zn^{2+} intercalating, here line scan for the intensity of the Se-Bi-Se columns in HAADF image are presented in Fig. R4f, from which we could obviously find that two Se atoms are symmetrically distributed at the side of Bi atom. As calculating the image contrast in intensity profile ($I_{\text{Bi}}/I_{\text{Se}}$) along the peach line of the HAADF image (Fig. R4f), an approximately proportional to the $(Z_{\text{Bi}}/Z_{\text{Se}})^{1.6}$ (Z is atomic number) confirms the ordered arrangement of Se and Bi atoms originating from the layered structure in the HAADF-STEM image (Fig. 1g). Hence the power law n in this experiment has been determined to be 1.6.

As comparing the line scan profiles of the HAADF images of E- Bi_2Se_3 and $\text{Zn}_x\text{Bi}_2\text{Se}_3$ in Fig. R5, we could find they are different apparently and some excess peaks site between two Se-Bi-Se columns in the $\text{Zn}_x\text{Bi}_2\text{Se}_3$ samples. In order to confirm the contribution of these peaks is from the insertion of Zn^{2+} , here we quantify the HAADF image by using the power law $n=1.6$. Firstly, we should point out that, if these peaks are contributed from the distortion of Se, the intensity profiles should be very similar with the ones in E- Bi_2Se_3 of Fig. R4d. Furthermore, the distance between the excess peak and the nearest Se-Bi-Se column neighbor is about 0.290 nm, which is almost no atoms or intensity here in the HAADF image of $\text{Zn}_x\text{Bi}_2\text{Se}_3$ sample. As using the power law of $n=1.6$ to calculate the average atomic number of Z' in the excess peaks, we could find that the average atomic number Z' is about 28, which is much smaller than the atomic number of Bi=83 and Se=53. Zn=30 indicating an unfulfilled Zn occupation inserted at the interval of two Se-Bi-Se columns (red spheres in Fig. R4c).

Please see Page 18 (Fig. 4) in the revised manuscript and Page 27 (Fig. S22) in the revised supplementary information.

Fig. R5 Comparison of the vertical line scan profiles of the HAADF images of E-Bi₂Se₃ in Fig. 1g and Zn_xBi₂Se₃ in Fig. 4j.

9. Sci Rep 8, 12325 (2018).

10. Ultramicroscopy 78, 111-124 (1999).

8. Some descriptions are very misleading. For example, “Even at -40°C, the capacity retentions remain to 106 % and 113 % compared to the capacities at 25 °C.” It should not be described as capacity retention because the final materials are changing with the decreased temperature.

Response:

We regret for any possible confusion which may have caused by our original statement. We change the expression to “Even at -40 °C, the capacities are still 1.06 (3 A g⁻¹) and 1.13 (1 A g⁻¹) times of the capacities at 25 °C.” This is a more precise expression.

9. All coulomb efficiency figures should be modified to a narrow percentage range.

Response:

We thank the referee for the valuable comment. All coulomb efficiency figures are modified to a narrow percentage range of 0-110 %. We are sorry to find that further narrowing the range will result in a mess between capacity data and coulomb efficiency in our figures.

Reviewer #2 (Remarks to the Author):

This manuscript by Zhao et al. reports an aqueous Zn/Bi₂Se₃ system in an antifreeze and concentrated electrolyte. Very surprisingly, in a temperature range of 25 - -20 °C, as the temperature decreases, the performance of the battery becomes even better. This has never been observed in any battery system: all battery will have capacity degradation at lower temperature. Bi₂Se₃ as a topological insulator contribute to the abnormal behavior. The Zn/ Bi₂Se₃ delivers high capacity and energy density at -20 °C (524 mAh g⁻¹ and 441 Wh kg⁻¹). Also, the mechanism explanations are solid with many characterizations to support. The improved performance is due to the enhanced contribution of surface topological electronic states and weaker lattice vibration of the few-layer topological insulator bismuth selenide at lower temperature. In general, this is a manuscript with high novelty. Hence, I recommend the manuscript to be published with minor revisions addressed.

Response:

Thank you for your recognition on our work. Highlights of our paper are listed as: As the temperature declines from 25 to -20 °C, the capacity of the Zn/E-Bi₂Se₃ battery is even remarkably increased. The unusual low-temperature electrochemical performance achieved with topological E-Bi₂Se₃ cathode is ascribed to the higher conductivity of E-Bi₂Se₃ at lower temperature resulting from the enhanced contribution of surface topological electronic states. In contrast, Zn/P-Bi₂Se₃ suffers a lot from sluggish Zn-transport kinetics owing to the rapid pulverization, low electrical conductivity and ion diffusion coefficient related to the finite interlayer spacing. Moreover, the electrical transport experiment demonstrates metallic temperature dependence for both of E-Bi₂Se₃ and discharging product Zn_xBi₂Se₃ with a prominently improved conductivity (roughly 1.2-1.5 times). Zn_xBi₂Se₃ exhibits higher conductivity than E-Bi₂Se₃ benefiting from the reinforced trivial metal states due to much-enlarged interlayer spacing and structure distortion after Zn²⁺ intercalating. In addition, we calculate the mean square displacement (MSD) of the topological E-Bi₂Se₃ to explore ion diffusion capability at lower temperature using MD calculation, concluding that the enhanced Zn ion diffusion capability in the topological E-Bi₂Se₃ at lower temperatures. This is attributed to the weaker lattice vibration of the material.

1. For the topological insulators, the surface states after exfoliating are enhanced and the electrical conductivity is also increased. Does this indicate that whether the material itself can be exfoliated is a key factor for the electrochemical performance in the selection of topological insulators?

Response:

We really appreciate the constructive comment from the referee. In the Bi_2Se_3 system, exfoliation enhances the topological state and battery performance. We also explore other three topological insulators in bulk, Sb_2Te_3 , $\text{Bi}_{0.9}\text{Sb}_{0.1}$ and Bi_2Te_3 (Fig. R6). We compare their low-temperature capacity retention properties $\left(\frac{\text{discharge capacity } (-20\text{ }^\circ\text{C})}{\text{discharge capacity } (20\text{ }^\circ\text{C})} \times 100\%\right)$ with common cathodes (MnO_2 and MoS_2) materials (Fig. R7). The results show that these battery capacities decrease with the decrease of temperature. With the same electrolyte, although the battery capacities at $-20\text{ }^\circ\text{C}$ are not better than their room-temperature capacities for Sb_2Te_3 , $\text{Bi}_{0.9}\text{Sb}_{0.1}$ and Bi_2Te_3 , their low-temperature capacity retention are much higher than that of MnO_2 and MoS_2 . It can be concluded that the capacity of topological electrode in bulk remains limited at low temperature. We further test MoS_2 electrode in the form of bulk and nanosheets. Obviously, the low-temperature capacity retention rate of nanosheets is higher than that of bulk (Fig. R8), which means the capacity at low temperature of common electrode can be improved to a certain extent after treatment like exfoliation comparing with the bulk. However, “better battery performance at lower temperature” is only observed in this exfoliated topological electrode Bi_2Se_3 .

Exfoliation endows Bi_2Se_3 with large electrode/electrolyte interfacial contact areas (reduce the ion diffusion length), good wettability and fast ion diffusion, which is helpful to improve the performance of batteries. Particularly, after exfoliating, the reduced concentration of bulk carriers in E- Bi_2Se_3 highlights the contribution of conductivity from surface topological states (Fig. S13, Page 15 in SI) significantly increasing electronic conductivity and boosting the electron transfer kinetics.^{11,12} In conclusion, exfoliation promotes the increase of active sites, and more importantly, it can enhance the topological protection performance of topological electrodes. What really gives the battery remarkably improved performance is the synergistic effect of exfoliation and exfoliation-enhanced topological surface state. Both topological properties and exfoliation contribute to the performance improvement of E- Bi_2Se_3 (25°C) vs. P- Bi_2Se_3 (25°C). In a word, the exfoliated (or nanostructured) topological insulators deliver better electrochemical performance than bulk because of the following two points: (1) Exfoliation endows Bi_2Se_3 with large electrode/electrolyte interfacial contact areas (reduce the ion diffusion length), good wettability and fast ion diffusion, which is helpful to improve the performance of batteries; (2)

Particularly, after exfoliating the reduced concentration of bulk carriers in E-Bi₂Se₃ highlights the contribution of conductivity from surface topological states significantly increasing electronic conductivity and boosting the electron transfer kinetics.

We are sorry for having not made this point clear. Please see highlighted parts on Page 12-13 in the supplementary information of the revised manuscript.

11. Nano Lett. 10, 3118-3122 (2010).

12. Nat. Mater. 9, 225-229 (2010).

Fig. R6 Discharge specific capacity versus temperature of three groups of topological cathodes: Zn//Sb₂Te₃ (a), Zn//Bi_{0.9}Sb_{0.1} (b), and Zn//Bi₂Te₃ (c) batteries at various current rates and temperatures in the HC-EGPAM hydrogel electrolyte.

Fig. R7 Comparison of capacity retention of topological cathodes (Sb₂Te₃, Bi_{0.9}Sb_{0.1}, and Bi₂Te₃) with conventional MoS₂ and MnO₂ cathodes at various current rates in the same HC-EGPAM

hydrogel electrolyte at -20°C. Here the capacity retention is calculated by $\left(\frac{\text{discharge capacity } (-20\text{ }^\circ\text{C})}{\text{discharge capacity } (20\text{ }^\circ\text{C})} \times 100\%\right)$.

Fig. R8 Comparison of capacity retention of MoS₂ bulk and MoS₂ nanosheets cathodes at -20 °C compared to that at 20 °C at various current rates.

2. On page 3, "...is formed by a periodic layers", and on page 4 "... the kinetics of the battery reactions at lower temperature.", and on page 8 "Thickness of numerous flakes is calculated...". Pay attention to the use of articles in the paper. Incorrect use of articles can also lead to confusion relating to singular vs plural senses.

Response:

We regret for this oversight and thank the referee for bringing them to our attention. These expressions have been corrected. Please see highlighted parts on Page 3, Page 4, Page 8 in the revised manuscript. We have also double checked our manuscript with special attentions on the use of articles.

3. The role of ethylene glycol in antifreeze electrolytes is critical. It is recommended to explain the role of ethylene glycol and the factors to consider in the selection of ethylene glycol content.

Response:

We thank the referee for the insightful comment. Low molecular vicinal alcohols, such as glycerol and ethylene glycol (EG), are well-known non-toxic inhibitors for water freezing and have been widely used as engine coolants in the industry.¹³ Pure EG freezes at about -12 °C.¹⁴ The freezing point of the EG/H₂O binary solution can be reduced below -40 °C by changing the EG concentration. In these binary solutions, alcohols form stable molecular clusters with H₂O molecules, which compete with hydrogen bonds in water therefore the saturated vapor pressure of water is significantly reduced, leading to a decrease of the saturated vapor pressure of water. As a result, the formation of crystal lattices of ice is disrupted and the freezing point is decreased.¹⁵

The freezing point is the main factor in our choice. 70% EG/water has the lowest freezing point of -55 °C (Table R1). In our high concentration of salts system, ethanol content can be as high as 50%, and when it is above 50%, the salts of LiTFSI and ZnTFSI can't completely dissolve. Thanks again for these practical and insightful questions.

Please find our revision for better clarify on Page 25 in the revised manuscript.

Table R1. Influence of EG to water ratios of coolant fluids on thermal, fluid-dynamic performance and heat transfer rate¹².

Fluids	Freezing point/°C	Boiling point/°C	Specific heat ^a /kJ kg ⁻¹ K ⁻¹	Cooling capacity/kW	Heat transfer coefficient/W m ⁻² K ⁻¹	Heat transfer rate/W ^b	Ref.
Pure EG	-13	197	2.5	n/a	n/a	n/a	3, 32
70%EG/water	-55	118	3.0	n/a	n/a	n/a	3, 32
50%EG/water	-34	107	3.4	7.2	152	18.4	305-308
30%EG/water	-15	104	3.8	8.5	189	n/a	305-308
Pure water	0	100	4.2	9.8	228	17.5	305-308

^a At 26.7 °C. ^b Total mass flux is 1244 kg m⁻² s⁻¹ for measuring heat transfer rate. Coolant fluid flow is 1000 kg h⁻¹ for measuring cooling capacity, pressure drop and overall heat transfer coefficient. n/a: not available.

13. *Angew. Chem. Int. Ed.* 56, 14159-14163 (2017).
14. *Chem. Soc. Rev.* 41, 4218-4244 (2012).
15. *Angew. Chem.* 130, 6678-6681 (2018).

4. In Fig 5g-j, when Zn²⁺ intercalated in Bi₂Se₃ interlayers (Zn²⁺Bi₂Se₃), is it still a topological insulator.

Response:

We thank the referee for this insightful comment. We conducted the electrical transport experiment with the sample of Zn^{2+} intercalated Bi_2Se_3 ($\text{Zn}^{2+}\text{Bi}_2\text{Se}_3$). Notably, the resistance of the discharging-product $\text{Zn}_x\text{Bi}_2\text{Se}_3$ nanosheets decreases as the temperature declines from 25 to -195 °C, in favorable agreement with the expected metallic behavior. Fig. 5g shows the calculated spin-orbit coupling (SOC) band structure of Zn^{2+} ions intercalated in Bi_2Se_3 interlayers ($\text{Zn}^{2+}\text{Bi}_2\text{Se}_3$), which gives a bandgap of 0.17 eV, illustrating the semiconducting nature of the bulk. The 6-QL slab models with a different type of elements terminated at the top and bottom surface are employed to investigate the surface electronic structure of the nanosheet (Fig. 5h-i). For all the Se-Se, Zn-Zn, and Se-Zn terminated slabs, the bandgap is closed and makes the system metallic, and the band structures do not show any noticeable difference. Compared with the bulk band structure, the conduction bands shift downward and mix with the valence bands, indicating the existence of metallic surface states. In addition, because Bi_2Se_3 itself has intrinsic topological surface states (Fig. S29b and S30a), even if the topological surface states still exist after Zn^{2+} have been embedded, the number of bands will be unchanged. The metallic surface states will play a leading role in the material's conductivity. Due to the overlapping of the conduction and valence bands, it is difficult to distinguish the topological states even if the topological surface states of $\text{Zn}^{2+}\text{Bi}_2\text{Se}_3$ are still retained.

We are sorry for having not made this point clear. For better clarity, we have revised our paper. Please see highlighted parts on Page 22 in the revised manuscript.

5. I think this paper may open a new venue for low temperature batteries, can the authors discuss more on what electrodes may have this unique feature other than topological insulators?

Response:

We thank the referee for this important question. In general, rechargeable ion battery performance should be degraded at a lower temperature compared to that at room temperature. Some materials with special physical and chemical properties at low temperatures have the opportunity to exhibit abnormal low-temperature properties. For example, topological semimetals with high electrical

conductivity and suitable carrier density near Fermi level are enticing candidate materials for electrochemical energy storage meriting from their topologically protected surface states.

We are sorry for having not made this point clear. We have revised our paper for better clarity. Please see highlighted parts on Page 23 in the revised manuscript.

Reviewer #3 (Remarks to the Author):

These authors report on the aqueous Zn//Bi₂Se₃ batteries in which the electrolyte is a high-salt/acrylamide gel medium and the cathode is an exfoliated form of Bi₂Se₃ previously reported to show improving conductivity as temperature is lowered. The paper is very detailed in terms of characterization, but the main findings are relatively straightforward: at lower temperature the batteries exhibit higher capacity than at ambient conditions. Authors attribute the effect mainly to materials physics & charge transport in the cathode, which seems reasonable considering that the control cells that use Bi₂Se₃ powder show mostly capacitive behaviors and minimal faradaic currents for the same electrolyte and temperature.

Overall, the study is of some interest, but there are at least three important concerns that would make it of limited technical impact. Although these concerns center on the battery performance and not the characterization, they are actually of greater importance because the origins of the improved charge transport in few layer Bi₂Se₃ are already known, so the materials themselves and the detailed analysis in the paper are actually of low novelty. The revised submission needs to clearly address the concerns below to allow this reviewer to understand what is the actual value of the contribution.

Response:

We thank you for recognition and we have provided new data to address your concerns. Please find our detailed response below.

Although there are many studies on Bi₂Se₃ as a topological insulator in the field of physics, this is the first time to use its topological features to achieve a better battery performance at lower temperature. This is important not only for application extension of Bi₂Se₃ materials, but also for low temperature batteries.

Major Concerns

1. The actual capacity of the cathodes used in the battery studies are impractically low (my estimate 0.4mAh/cm²), which would make such Zn batteries not competitive with state-of-the-art commercial Zn batteries and impractical relative to Li-ion batteries, regardless of their good low temperature behaviors. Considering that the EIS results and previous literature indicate that E-Bi₂Se₃ becomes more conductive at low temperature, it would seem that cathode capacities at least one order of magnitude higher (e.g. 2-4mAh/cm²) would be viable. Yet authors do not report results for such systems. Please explain how the cathode capacity was selected? Please also provide results at higher cathode capacity (at least 2mAh/cm²).

Response:

We thank the referee for this insightful comment. E-Bi₂Se₃ cathodes with a mass loading of 1-4 mg cm⁻² were prepared and tested at current densities of 0.1 A/g to 5 A/g (Fig. R9 and Fig. S14). Fig. R9a and Fig. S14 plot the areal capacity vs. the areal mass loading of the electrodes at -20 ° C. The areal capacity increases linearly with the increasing areal mass loading when the current densities are less than 3 A g⁻¹. At a higher current density, the dependence slightly deviates from the linear relationship. Fig. R9b shows the areal capacity as a function of the areal current density in which the E-Bi₂Se₃ cathode with a mass loading of 4 mg cm⁻² exhibits areal capacities of 2.1, 1.84, 1.52, 1.17, 0.72, and 0.46 mAh cm⁻² at the areal current density of 0.4 (0.1 A/g), 1.2 (0.3 A/g), 2 (0.5 A/g), 4 (1 A/g), 12 (3 A/g), and 20 (5 A/g) mA cm⁻², respectively. The temperature-dependent areal capacity of E-Bi₂Se₃ cathodes and the areal capacity vs. areal mass loading at 0.3 A/g are displayed in Fig. R9c and Fig. R9d, respectively. **The E-Bi₂Se₃ cathode achieves an areal capacity of 2.1 mAh cm⁻² with a mass loading of 4 mg cm⁻² at 0.1 A/g and -20 ° C.**

We do understand your concern on practical application of the developed batteries. On the one hand, we need to rely on the current progress on the whole Zn battery community, which is behind of lithium-ion batteries in terms of technology maturity. On the other hand, the developed Zn//E-Bi₂Se₃ batteries possess state-of-art performance in comparison with other Zn batteries, especially at low temperatures (Fig. S16b). More important, at the current stage of fundamental study, we demonstrate a unique phenomenon that a better battery performance can be achieved at lower temperature, which may find applications in the scenarios that batteries are used long-termly in cold environments.

We will keep your comments in mind and keep improving our battery in terms of the loading mass

and cost of electrolyte for potential practical applications. When the battery can be eventually used, probably the cathode and electrolyte are both different from current materials, but the idea and concept are from the current reported system.

We are sorry for having not made this point clear. Please see Page 14 in the revised manuscript and Page 16 (Fig. S14) in the revised supplementary information.

Fig. R9 Zn//E-Bi₂Se₃ batteries with different cathode mass loadings. (a) The areal capacity of E-Bi₂Se₃ cathodes with mass loading of 1-4 mg cm⁻² from 0.1 A/g to 5 A/g at -20 °C. (b) The areal capacity vs. the areal current density of E-Bi₂Se₃ cathodes with a mass loading of 1-4 mg cm⁻² at -20 °C, respectively. (c) Temperature-dependent areal capacity of E-Bi₂Se₃ cathodes with mass loading of 1-4 mg cm⁻² at 0.3 A/g. (d) The areal capacity of E-Bi₂Se₃ cathodes with mass loading of 1-4 mg cm⁻² from -20 °C to 20 °C at 0.3 A/g.

2. The high concentration of LiTFSI (21 m) in the electrolytes is also impractically high. Again, such batteries would not be of practical interest or competitive with commercial alkali Zn cells.

Considering that the authors also use an antifreeze in the electrolyte, is the high salt concentration really necessary. It will be helpful to the readers to understand how the antifreeze to salt ration influence the low temperature ion transport behaviors in the HC-EGPAM electrolytes. In other word, if as the authors claim, the higher low temperature capacity comes from enhance transport in the cathode, why is so much salt needed in electrolyte?

Response:

Thank you for your insightful comments. Our initial intention is to explore novel electrode materials to provide more possibilities for low-temperature batteries. In this case, we have to base our idea on a collaborative design of the electrolyte and electrode to obtain the high-performance low-temperature battery. To obtain superior low-temperature performance, it is very critical and essential to ensure that the electrolyte is in a good working condition at low temperature.

That is, we need an electrolyte with excellent low temperature performance as a platform to study and manifest the unique features of the developed topological insulating E-Bi₂Se₃ electrode.

In the selection of electrolytes, we first consider the antifreeze electrolyte at low temperatures. However, the temperature of our battery is expected to work at a temperature as low as -50 °C. The freezing point of 2 M ZnSO₄ electrolyte with the 60% EG-to-water volume ratio is only -33 °C.¹⁶ Therefore, the low-temperature antifreeze is not good enough for such a low temperature. So we choose the electrolyte of “EG/water in salt” (1 m Zn(TFSI)₂ + 21 m LiTFSI (where m is molality (mol kg⁻¹)).¹⁶ Even at -50°, a conductivity of 0.035 ms cm⁻¹ can be still maintained, which is comparable to or even better than many other reported low-temperature electrolytes. Therefore, we believe the HC-EGPAM will not drag down the battery performance much at low temperature, which make it possible to manifest the unique performance and contribution of the developed electrode materials, that is, E-Bi₂Se₃. We supplement the comparison of the ionic conductivity of HC-EGPAM and HC-PAM with various other low-temperature electrolytes in Fig. S9k.¹⁷⁻²⁰

Zn//E-Bi₂Se₃ batteries show better performance at low temperatures in the HC-EGPAM electrolyte, but Zn//P-Bi₂Se₃ does not show better performance at low temperatures in the same electrolyte. This indicates that the better performance at low temperature is the synergistic effect of the specific cathode (E-Bi₂Se₃) and HC-EGPAM electrolyte. In fact, if only the low-temperature transport of the cathode is enhanced but the electrolyte delivers a sharp drop in ion conductivity at a low

temperature, we cannot expect a good low-temperature performance of the battery. It is clear that we cannot rely on electrolytes or electrodes alone for better performance at low temperatures. Here HC-EGPAM provide a good platform for a battery to work at low temperature and E-Bi₂Se₃ is the key to achieve a better performance at low temperature.

The high concentration of LiTFSI (21 M) in the electrolytes will inevitably increase the cost of the battery. But the main idea here is an approach to achieve a better performance at lower temperature. Following this concept, further optimization to electrolytes, including reducing their cost, will be further investigated. This may take some time but the concept is come from our paper.

We thank the referee for this insightful comment. We have added related discussion in the revised manuscript (Page 23).

16. Nat. Mater. 17, 543-549 (2018).
17. Angew. Chem. Int. Ed. 131, 17150-17155 (2019).
18. *Joule* 2, 902-913 (2018).
19. Chem. Commun. 56, 9640-9643 (2020).
20. Energy Stor. Mater. 9, 47-69 (2017)

3. In the abstract and at several places in the body of the manuscript, authors mention that their Zn//E-Bi₂Se₃ batteries can operate to temperatures as low as -50 C °. It would be helpful to see the ionic conductivity data for the HC-EGPAM the electrolytes (Fig. S9) than those reported now.

Response:

Thanks for the insightful comment. We supplement the ionic conductivity data of the HC-EGPAM at lower temperatures to -50 °C in Fig. S9f (Page 9 in the revised supplementary information). We further compare the ionic conductivity of HC-EGPAM and HC-PAM with various other low-temperature electrolytes as shown in Fig. S9k. The HC-EGPAM hydrogel electrolyte shows a superior ionic conductivity comparable to DMSO-water electrolyte, ethyl acetate-based cosolvent electrolyte, carbonate-based electrolyte, and ionic liquid electrolyte (Figure. S9k).

We thank the referee for this insightful comment. We have added related discussion in the revised supplementary information (Page 11 and Page 12).

REVIEWER COMMENTS

Reviewer #1 (Remarks to the Author):

In the revised version, the authors address the majority of my comments/concerns, but still with a few not well clarified, particularly the ones regarding the Zn insertion mechanism. More analyses and discussions are needed as following:

1. In the response to my Comment 3, in the revised version, the authors changed their claim of "firstly observed" to "rarely observed", which is fine. But since they discussed the differences between this work and theirs, why not include such discussions in the revised manuscript? The novelty and significance of this work compared to the previous ones should deserve a detailed clarification, rather than just changing a word.

2. In response to my comment 4, the authors mentioned that "atomic configuration in the HAADF image could be identified clearly...in Figure R3". But from the given image, how could the authors make this claim?

3. EDS mappings in Figure. S23 cannot justify the uniform distribution of Zn-Bi-Se since it is a low-mag SEM-EDS, which might involve signals from electrolyte salts. In addition, considering the fake mapping signals from elemental analysis that is well known in EDS analysis, an EDS spectrum from the same mapping data set with the element peaks clearly identified should be given to confirm their presence. More importantly, in the EELS mapping in Figure 4K, the Zn, again, shows a different distribution to that of Bi and Se, where it seems that Zn is only on the surface region. This raises a serious concern that Zn intercalation into the particle is not the actual situation and that Zn is only absorbed on the particle surface (capacitive behavior?). Besides, this region is too small to represent the whole particle. I suggest the authors to do an EELS-mapping of one whole particle.

Reviewer #1 (Remarks to the Author):

In the revised version, the authors address the majority of my comments/concerns, but still with a few not well clarified, particularly the ones regarding the Zn insertion mechanism. More analyses and discussions are needed as following:

1. In the response to my Comment 3, in the revised version, the authors changed their claim of "firstly observed" to "rarely observed", which is fine. But since they discussed the differences between this work and theirs, why not include such discussions in the revised manuscript? The novelty and significance of this work compared to the previous ones should deserve a detailed clarification, rather than just changing a word.

Answer:

We thank the referee for this insightful comment. We have revised our manuscript to clarify this point. We have revised our statement as follows:

"This abnormal low-temperature response of battery performance has also been observed in a $\text{Na}_3\text{VCr}(\text{PO}_4)_3$ (NVCP) cathode. However, it should be noted that this is because NVCP is not stable at a relatively high temperature (30°C), leading to a poorer performance output at higher temperature. At low temperature, the high temperature induced irreversible electrochemical reaction can be suppressed in NVCP¹. With a different mechanism, here exfoliation endows Bi_2Se_3 with large electrode/electrolyte interfacial contact areas (reduced the ion diffusion length) and fast ion diffusion, which improves the performance of batteries. More importantly, exfoliation enhances the topological protection performance of the electrodes, significantly increasing electronic conductivity and boosting the electron transfer kinetics ... "

Please find the highlighted revision in the revised manuscript (Page 13 and 14).

1. Adv. Energy Mater. 11, 2003256 (2021).

2. In response to my comment 4, the authors mentioned that "atomic configuration in the HAADF image could be identified clearly...in FIgure R3". But from the given image, how could the authors make this claim?

Answer:

We thank the reviewer for the insightful comments and sorry for the

inappropriate expression. Indeed, while the detailed line profile (Figure S5f) can be used to identify Se and Bi atoms, the image does not reveal the atomic configuration. Following your comment, we have revised our statement as follows in our revised version:

“However, although the HAADF image (Figure S5e) does not clearly reveal the atomic configuration, as fitting the peak intensity of line profile in Figure. S5f, we can still see two Se atoms do not distribute symmetrically around Bi site (the distance of Se-Bi at the left side is 0.248 nm, and the distance of Bi-Se at the right side is 0.171 nm, which is an obvious difference as considering the experimental error of 0.03 nm). Normally, ball-milling, and hydrothermal stripping could affect the crystallinity greatly during the sample preparation. Especially, it is well-known that the electron beam knock-on may displace the atoms in crystal.² When we tried to capture the high resolution HAADF image along [010] zone axis, the atomic configuration at different areas may have certain distortion. This can explain the observed unsymmetrical Bi atom sitting between two Se atoms.”

Once again, we thank the referee for this comment. We have revised the explanation in the revised supplementary information. Please see highlighted parts on Page 8 in the SI.

3. EDS mappings in Figure. S23 cannot justify the uniform distribution of Zn-Bi-Se since it is a low-mag SEM-EDS, which might involve signals from electrolyte salts. In addition, considering the fake mapping signals from elemental analysis that is well known in EDS analysis, an EDS spectrum from the same mapping data set with the element peaks clearly identified should be given to confirm their presence. More importantly, in the EELS mapping in Figure 4K, the Zn, again, shows a different distribution to that of Bi and Se, where it seems that Zn is only on the surface region. This raises a serious concern that Zn intercalation into the particle is not the actual situation and that Zn is only absorbed on the particle surface (capacitive behavior?). Besides, this region is too small to represent the whole particle. I suggest the authors to do an EELS-mapping of one whole particle.

Answer:

Thank you for your insightful comments. We totally agree that firm evidences should be provided to prove the Zn ions intercalation to the lattice. In the XRD patterns of the E-Bi₂Se₃ cathode at selected charge/discharge states (Fig. 4a and 4b), the offset of the peak position indicates that intercalation/de-intercalation of Zn²⁺ triggers expansion/reversible contraction of the interlayer

spacing. *Ex situ* Raman spectra of the three characteristic peaks of the E-Bi₂Se₃ cathode at selected charge/discharge states also show changed peak width and peak height corresponding to the embedding and deintercalation of Zn²⁺ into the interlayers of E-Bi₂Se₃ (Fig. 4c). More importantly, the HAADF-STEM images of Zn_xBi₂Se₃ in Fig. 4i and 4j reveal Zn²⁺ occupies the position between the Se layers. All these results prove that Zn²⁺ is embedded in the E-Bi₂Se₃ rather than adsorbed on the surface.

Following your suggestion, as EELS spectrum is more sensitive to elements with light atomic mass, here, we provide high-magnification TEM-EDS mappings of one whole particle of fully discharged E-Bi₂Se₃ (Fig. R1(a-d)), and the EDS spectrum from the same mapping data set with the element peaks clearly identified is given to confirm the presence of Bi, Se and Zn (Fig. R1(e)). The elemental distribution reveals that Zn is uniformly distributed in the E-Bi₂Se₃.

We are sorry for having not made this point clear. Please see new Fig. 4k and related description on Page 18 in the revised manuscript and related EDS spectrum (new Figure. S23) on Page 29 in the revised supplementary information.

Fig. R1 (a-d) TEM dark field image of fully discharged E-Bi₂Se₃ and corresponding EDS elemental mapping of Bi, Se, and Zn. (e) Corresponding EDS spectrum of the fully discharged E-Bi₂Se₃.

4. Referee #1 suggests provide a comparison between the HC-EGPAM of this work with other EGPAM already reported in the literature in order to rule out the contribution from the electrolyte.

Answer:

Thanks for your insightful comments. To rule out the contribution from the electrolyte, we compare MnO₂//Zn batteries using HC-EGPAM of this work with other antifreeze hydrogel electrolytes already reported (Fig. R2a). Capacities of MnO₂//Zn batteries using HC-EGPAM and other antifreeze hydrogel electrolytes all decrease with decreasing temperature, and MnO₂//Zn batteries using HC-EGPAM and EGPAM electrolytes show similar capacity retention rates at different temperatures. This phenomenon indicates that our HC-EGPAM has a similar anti-

freezing ability with reported electrolytes.

Then, we compare our topological insulator E-Bi₂Se₃ electrode with Prussian blue analogue (FeHCF), V₂O₅, MnO₂ and Co₃Sn_{1.8}S₂ electrodes in the same HC-EGPAM (Fig. R2b). Considering that the performance of other electrode materials remarkably decreases with the decrease of temperature, we confirm that the exfoliation-enhanced topological surface state of the E-Bi₂Se₃ is the key to achieve the improved performance at low temperature. Of course, to obtain superior low-temperature performance, it is essential to ensure that the electrolyte is in a good working condition at low temperature.

We have revised our paper. Please see highlighted parts on Page 13 in the revised manuscript and Page 15 (Fig. S12b and c) in the revised supplementary information.

Fig. R2 (a) Temperature-capacity retention dependence of MnO₂//Zn batteries using HC-EGPAM of this work with other MnO₂//Zn batteries reported using other antifreeze hydrogel electrolytes.^{2,3} Here an antifreeze polyacrylamide (PAM) hydrogel electrolyte with highly concentrated salts incorporated along with ethylene glycol is denoted as HC-EGPAM. PAM hydrogel electrolyte with 2M ZnSO₄ salt incorporated along with ethylene glycol is denoted as EGPAM. CT3G30 hydrogel electrolyte uses cotton as the raw material, tetraethyl orthosilicate as the crosslinker, and glycerol as the antifreezing agent, where C, T, 3, G, and 30 represent cellulose, tetraethyl orthosilicate (TEOS), the milliliters of TEOS added, glycerol, and the milliliters of glycerol added, respectively. (b) Temperature-capacity retention dependence of E-Bi₂Se₃ and other cathodes using the same HC-EGPAM electrolyte.⁴

2. Energy Environ. Sci. 12, 706–715 (2019).
3. Adv. Mater. 33, 2007559 (2021).
4. Angew. Chem. Int. Ed. doi:10.1002/ange.202111826 (2021).

.....,

Once again, thank you very much for your comments and suggestions.

Yours Sincerely,

Chunyi ZHI

REVIEWERS' COMMENTS

Reviewer #1 (Remarks to the Author):

The authors completely addressed my comments this time. I am happy to give a green light for the MS to be published in Nature Commun.